

# 3d large $N$ vector models at the boundary

**Lorenzo Di Pietro[1,2⋆], Edoardo Lauria[3†] and Pierluigi Niro[4,5‡]**

**1** Dipartimento di Fisica, Università di Trieste, Strada Costiera 11, I-34151 Trieste, Italy
**2** INFN, Sezione di Trieste, Via Valerio 2, I-34127 Trieste, Italy
**3** CPHT, CNRS, Institut Polytechnique de Paris, France
**4** Physique Théorique et Mathématique and International Solvay Institutes,
Université Libre de Bruxelles, C.P. 231, 1050 Brussels, Belgium
**5** Theoretische Natuurkunde, Vrije Universiteit Brussel,
Pleinlaan 2, 1050 Brussels, Belgium

⋆ ldipietro@units.it, † edoardo.lauria@polytechnique.edu, ‡ pierluigi.niro@ulb.ac.be

## Abstract

We consider a 4d scalar field coupled to large $N$ free or critical $O(N)$ vector models, either bosonic or fermionic, on a 3d boundary. We compute the $\beta$ function of the classically marginal bulk/boundary interaction at the first non-trivial order in the large $N$ expansion and exactly in the coupling. Starting with the free (critical) vector model at weak coupling, we find a fixed point at infinite coupling in which the boundary theory is the critical (free) vector model and the bulk decouples. We show that a strong/weak duality relates one description of the renormalization group flow to another one in which the free and the critical vector models are exchanged. We then consider the theory with an additional Maxwell field in the bulk, which also gives decoupling limits with gauged vector models on the boundary.

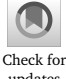

# 1 Introduction

The coupling between two quantum field theories (QFTs) that are defined in different space-time dimensions is an interesting probe of the dynamics of both theories. On the one hand, these couplings define a set of extended objects in the higher-dimensional "bulk" theory, either boundaries or defects, that enlarge the observables beyond the correlation functions of local operators. On the other hand, from the point of view of the QFT localized on the lower-dimensional boundary or defect, these couplings can be interpreted as a very special set of non-local interactions, that lead to non-standard renormalization group (RG) flows and fixed points. Examples of this are the construction of the long-range Ising model as a theory living on a defect (though one of non-integer codimension) in [1], that shed light on the conformal

symmetry of the theory, and the generalizations considered in [2].

At special points in the parameter space the bulk and the boundary might decouple, defining a new local theory on the boundary or defect, that might be harder to reach from the original theory if one is restricted to local interactions. This phenomenon occurs for example when coupling a 4d Maxwell field to a 3d conformal field theory (CFT) with a $U(1)$ global symmetry. As shown in [3], this theory features infinitely many bulk/boundary decoupling limits in which the boundary theory is the image of the original 3d CFT under the $SL(2,\mathbb{Z})$ action of [4], and generically contains 3d abelian gauge fields.

In this paper we study another instance of the connection between two local CFTs through the coupling with a higher-dimensional theory. As in the example mentioned above, the theories that we consider live on the 3d boundary of a 4d space, with a free field in the bulk, though this time a single massless scalar. Before turning on any interaction on the boundary, the possible conformal boundary conditions are either Dirichlet, with a boundary mode $\partial_\perp \Phi$ of scaling dimension 2, or Neumann, with a boundary mode $\Phi$ of dimension 1. The 3d theories that we put on the boundary are the vector models, i.e. theories of $N$ scalar fields $\varphi^I$ or alternatively spin-1/2 fields $\psi^I$, with $I = 1,\ldots,N$, where "vector" refers to the fields being in the fundamental representation of an $O(N)$ global symmetry. We consider either the free (massless) vector models, or the so-called "critical" ones, i.e. the 3d CFTs that can be reached from the free ones through an RG flow via a quartic $O(N)$-invariant interaction. This interacting CFT is typically referred to as $O(N)$ model in the bosonic case, and Gross-Neveu CFT in the fermionic case. As is well-known, the critical vector models can be solved in the limit $N \to \infty$ and are amenable to a $1/N$ perturbation theory. In particular, at the leading order at large $N$ they differ from the free vector models only by a Legendre transform, whose effect is to replace the $O(N)$-invariant quadratic scalar operator $\varphi^I \varphi^I$ of dimension 1 ($\bar{\psi}^I \psi^I$ of dimension 2) by an operator $\sigma$ of scaling dimension $2 + \mathcal{O}(N^{-1})$ $(1 + \mathcal{O}(N^{-1})$, respectively).

As a result, we can form the following classically marginal couplings between the free scalar in the bulk and the vector models on the boundary. For the bosonic vector model:

$$\text{Dirichlet} + N \text{ free scalars} + g \int d^3x\, \partial_\perp \Phi\, \frac{\varphi^I \varphi^I}{\sqrt{N}}\,,$$
$$\text{or}$$
$$\text{Neumann} + N \text{ critical scalars} + g' \int d^3x\, \Phi\, \sigma\,.$$

Similarly for the fermionic model:

$$\text{Neumann} + N \text{ free fermions} + g \int d^3x\, \Phi\, \frac{\bar{\psi}^I \psi^I}{\sqrt{N}}\,,$$
$$\text{or}$$
$$\text{Dirichlet} + N \text{ critical fermions} + g' \int d^3x\, \partial_\perp \Phi\, \sigma\,.$$

Both these couples of theories can be solved in the limit of large $N$ with any fixed value of $g$ and $g'$, not necessarily small.[1] At the leading order at large $N$ the operator from the 3d sector is a generalized free field and therefore there is a line of fixed points parametrized by $g$ or $g'$ [9]. These fixed points are lifted at the subleading order, namely at order $\mathcal{O}(N^{-1})$. We obtain the $\beta$ functions for these couplings at order $\mathcal{O}(N^{-1})$ and to all orders in the couplings $g$ and $g'$. We find that each of the $\beta$ functions vanishes when the corresponding coupling is either 0 or $\infty$. In the bosonic theory, one also needs to consider the running of the sextic $O(N)$-invariant coupling $(\varphi^I \varphi^I)^3$ but it is still possible to find a real fixed point when $g$ and $g'$

---

[1]Small $N$ versions of some of these theories appeared before in the literature. Ref. [5] considered the free bosonic theory for $N = 1$ and the free fermionic theory for $N = 2$ under the name of "mixed scalar" and "mixed Yukawa" theories. Ref.s [6–8] considered the deformation of the 3d Ising CFT by the product of the energy operator and a generalized free field. With the appropriate choice of the scaling dimension (corresponding to $s = -1$) this can be seen as the $N = 1$ version of our critical scalar theory.

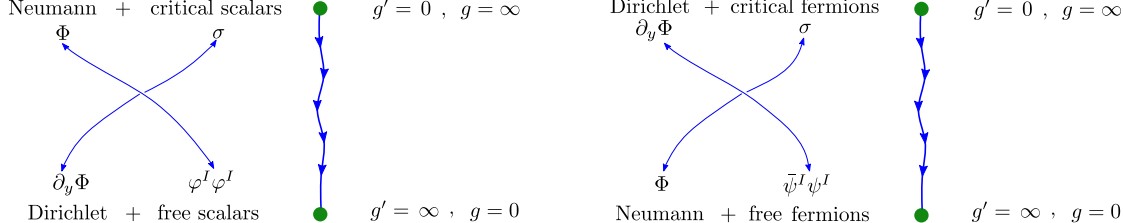

Figure 1: The bosonic (left) and fermionic (right) boundary RG flows at large $N$. Each admits two dual descriptions, as a marginally relevant deformation with coupling $g'$ of the UV fixed point or as a marginally irrelevant deformation with coupling $g$ of the IR fixed point. The duality maps $g'$ to $\pm 1/g$, and maps operators according to the arrows.

are either 0 or $\infty$. Moreover, both in the bosonic and the fermionic theory the $\beta$ functions of $g$ and $g'$ and the anomalous dimensions of the $O(N)$-vector fields are mapped into each other by setting $g' = \pm 1/g$ (there is a relative sign between the bosonic and the fermionic case).

We show that the theories with couplings $g$ and $g' = \pm 1/g$ are in fact dual descriptions of the same RG flow. This can be seen as an extension to order $\mathcal{O}(N^{-1})$ of the duality acting on the line of fixed points of [9]. We provide a simple path-integral argument at large $N$ for the duality, based on rewriting both the $g$ and the $g'$ deformations in terms of a non-local quartic interaction between the $O(N)$-vector fields, which is conveniently written in momentum space as

$$\int \frac{d^3 p}{(2\pi)^3} |p| (\varphi^I \varphi^I)(p)(\varphi^J \varphi^J)(-p) \quad \text{and} \quad \int \frac{d^3 p}{(2\pi)^3} \frac{1}{|p|} (\bar{\psi}^I \psi^I)(p)(\bar{\psi}^J \psi^J)(-p) \,,$$

in the bosonic and in the fermionic theory, respectively. Non-local generalizations of bosonic vector models were also considered in [2, 10–12] with the difference that the non-locality resides in the kinetic term of the vector fields rather than in their quartic interaction. Similar strong/weak dualities were also found in [13, 14] for purely three-dimensional RG flows involving couplings between multiple vector models.

While the bulk scalar and the 3d degrees of freedom are coupled along the RG flow, both in the UV and in the IR fixed points they decouple. At one end of the RG the local theory on the boundary is the free vector model, and at the other end one instead finds the critical vector model. The two resulting RG flows are depicted in fig. 1. The fact that such a decoupling happens when we take the bulk/boundary coupling to be strong would look surprising, if we did not have a strong/weak duality to explain it. As a result, these fixed points do not define interacting conformal boundary conditions for the 4d free scalar CFT, in qualitative agreement with the tight restrictions that were recently found with the numerical conformal bootstrap [15]. We also check the monotonicity along boundary RG flows of the hemisphere partition function [16, 17] comparing the values at these decoupled UV/IR fixed points.

We then consider adding an additional Maxwell field in the bulk, with Neumann boundary condition, whose boundary value is coupled to the current of a $U(1)$ subgroup of the $O(N)$ symmetry acting on the boundary degrees of freedom. From the point of view of the 3d theory this gives rise to an additional non-local interaction parametrized by the bulk gauge coupling, that unlike $g$ is exactly marginal to all orders in $1/N$. Going to large values of the gauge coupling, one then finds new bulk/boundary decoupling limits in which the local 3d sector is the $U(1)$-gauged version of those mentioned above, namely bosonic and fermionic $QED_3$ with a large number of flavors, with or without a quartic interaction at criticality. In this more general theory there are also families of fixed points in which the value of $g$ depends on the gauge coupling $\lambda$. As $\lambda$ is dialed to be large, they annihilate in pairs and become complex

before reaching the decoupling limit for the bulk gauge field $\lambda \to \infty$. Therefore also in this case we do not find an example of a unitary and interacting conformal boundary condition for the bulk free scalar. However, we do find examples if we also allow a bulk $\theta$ term, as we discuss in the appendix B.

## 2 Large $N$ scalars on the boundary

### 2.1 Dirichlet coupled to $N$ free scalars

We start by analyzing the theory of the free bulk scalar field $\Phi$ with Dirichlet boundary condition coupled to the free bosonic vector model, i.e. $N$ 3d free scalars $\varphi^I$. We denote the coordinates on the boundary with $x^a$, $a = 1, 2, 3$, and the coordinate perpendicular to the boundary with $y \geq 0$. The index for the bulk coordinates will be denoted with $\mu = 1, \ldots, 4$. The $O(N)$-invariant coupling between the two sectors can be written as follows

$$S_{\text{D}}^b[g,h] = \int_{y \geq 0} d^3x \, dy \, \frac{1}{2}(\partial_\mu \Phi)^2 + \int_{y=0} d^3x \left[ \frac{1}{2}\left(\partial_a \varphi^I\right)^2 + \frac{g}{\sqrt{N}} \partial_y \Phi \, \varphi^I \varphi^I + \frac{h}{N^2}(\varphi^I \varphi^I)^3 \right] .$$

(2.1)

All the relevant couplings such as $\varphi^I \varphi^I$, $(\varphi^I \varphi^I)^2$ are fine-tuned to zero. We are left with the cubic and sextic couplings of (2.1), which are the only classically marginal interactions preserving $O(N)$. The particular scaling with $N$ that we have chosen is such that the theory admits a non-trivial limit of $N \to \infty$ with $g$ and $h$ fixed and not necessarily small, that allows a systematic $1/N$ expansion. The Feynman rules are given in appendix A. Note that the boundary action breaks the $\mathbb{Z}_2$ symmetry $\Phi \to -\Phi$ of the bulk theory, which acts on the couplings as $(g, h) \to (-g, h)$. The boundary interaction gives rise to a "modified Dirichlet" boundary condition[2]

$$\Phi|_{y=0} = -\frac{g}{\sqrt{N}} \varphi^I \varphi^I .$$

(2.2)

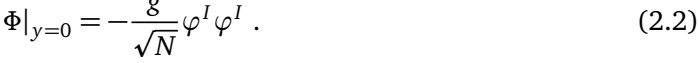

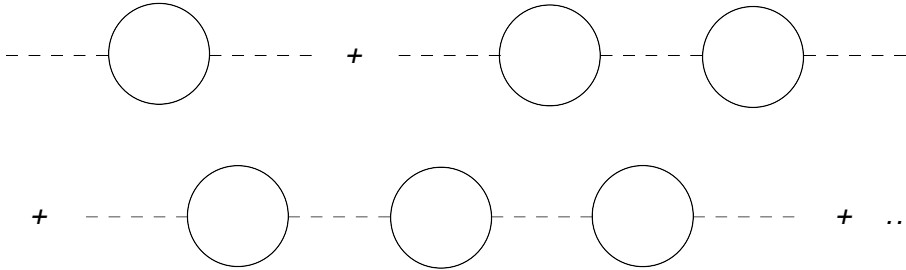

Figure 2: Diagrams that contribute to the boundary propagator of $\partial_y \Phi$ in the limit of large $N$ with $g$ fixed.

As a first step, we need to compute the propagators of $\partial_y \Phi$ between two points on the boundary. At the leading order at large $N$ this propagator receives corrections from bubble diagrams connected by tree-level propagators of $\partial_y \Phi$, see fig. 2. Going to boundary momentum variables, these corrections lead to a geometric series that can be easily resummed. Since the bubble of a scalar field in 3d gives $(4|p|)^{-1}$ and the tree-level boundary propagator of $\partial_y \Phi$ is $-|p|$, the corrections are actually independent of momentum and they only give a $g$-dependent

---

[2]We are implicitly adding a boundary term $\int_{y=0} d^3x \, \Phi \partial_y \Phi$, so that the total variation of the quadratic action for the scalar field is $\int_{y=0} d^3x \, \Phi \delta(\partial_y \Phi)$.

normalization of the two-point function. In a similar fashion we can obtain the two-point function of the operator $\frac{1}{\sqrt{N}}\varphi^I\varphi^I$ and the mixed one. The geometric sums give

$$\langle\partial_y\Phi(p)\partial_y\Phi(-p)\rangle = \frac{1}{1+g^2/4}(-|p|)\,,$$
$$\langle\frac{1}{\sqrt{N}}\varphi^I\varphi^I(p)\frac{1}{\sqrt{N}}\varphi^J\varphi^J(-p)\rangle = \frac{1}{1+g^2/4}\frac{1}{4|p|}\,, \tag{2.3}$$
$$\langle\partial_y\Phi(p)\frac{1}{\sqrt{N}}\varphi^I\varphi^I(-p)\rangle = \frac{g/4}{1+g^2/4}\,.$$

This is the only effect of the interaction at the leading order at large $N$. In particular, $g$ and $h$ have vanishing $\beta$ functions in the strict $N\to\infty$ limit and give rise to a two-parameter family of fixed points. This family of fixed points is however only approximate, and it gets lifted by non-trivial $\beta$ functions at the subleading order in $1/N$.

### 2.1.1 Beta functions and anomalous dimension

We now use the propagator (2.3) to compute the $1/N$-suppressed perturbative corrections. The calculation of the RG functions at leading non-trivial order in the $1/N$ expansion is akin to a one-loop calculation in ordinary perturbation theory, in that it amounts to extract a universal logarithmic divergence and therefore it is not sensitive to the choice of regularization. For definiteness, we will adopt a Wilsonian approach and use a hard cutoff $\Lambda$ on the boundary momenta running in loops. There is no need to renormalize the bulk field or the bulk action, because the interactions are localized on the boundary and therefore by locality only boundary couplings can run. Using a subscript $\Lambda$ to denote quantities relative to the theory with a certain UV cutoff, after an RG step in which we integrate out a shell of momenta between $\Lambda$ and $\Lambda' < \Lambda$, we relate the variables as

$$\varphi^I_{\Lambda'} = Z_\varphi^{1/2}\varphi^I_\Lambda\,, \quad g_{\Lambda'} = Z_\varphi^{-1}Z_g g_\Lambda\,, \quad h_{\Lambda'} = Z_\varphi^{-3}Z_h h_\Lambda\,. \tag{2.4}$$

The diagrams that contribute to the wave function renormalization of $\varphi^I$ and to the renormalization of the $g$ and $h$ vertices are depicted in fig. 3. The renormalization constants are

$$\delta Z_\varphi = \frac{2}{3\pi^2 N}\frac{g^2}{1+g^2/4}\log(\Lambda/\Lambda')\,, \quad \delta Z_g g = -\frac{2}{\pi^2 N}\frac{g^3}{1+g^2/4}\log(\Lambda/\Lambda')\,,$$
$$\delta Z_h h = \frac{1}{\pi^2 N}\frac{9h^3/32 - 9h^2 - 3\left(g^4/8 + g^2 + 10\right)g^2 h + 16g^6/3}{(1+g^2/4)^3}\log(\Lambda/\Lambda')\,, \tag{2.5}$$

where $Z_{(\cdot)} = 1 + \delta Z_{(\cdot)}$ up to subleading corrections at large $N$. We obtain the following results for the anomalous dimension $\gamma_\varphi$ of the vector field and for the $\beta$ function of the coupling $g$

$$\gamma_\varphi = \frac{d\log Z_\varphi^{1/2}}{d\log\Lambda} = \frac{1}{3\pi^2 N}\frac{g^2}{1+g^2/4} + \mathcal{O}(N^{-2})\,,$$
$$\beta_g = -\frac{d}{d\log\Lambda}(Z_\varphi^{-1}Z_g g) = \frac{8}{3\pi^2 N}\frac{g^3}{1+g^2/4} + \mathcal{O}(N^{-2})\,. \tag{2.6}$$

In the appendix C we check these results with a calculation in dimensional regularization. We see that the coupling $g$ is *marginally irrelevant* in the vicinity of the decoupled point $g = 0$, which therefore is IR stable. There is another zero of $\beta_g$ at $g = \infty$, around which $g$ is marginally relevant, i.e. it is a UV stable fixed point. This is more evident using the "compactified" variable $f_g = \frac{g^2/4}{1+g^2/4} \in [0,1]$, in terms of which the $\beta$ function becomes

$$\beta_{f_g} = \frac{64}{3\pi^2 N}f_g^2(1-f_g)\,, \tag{2.7}$$

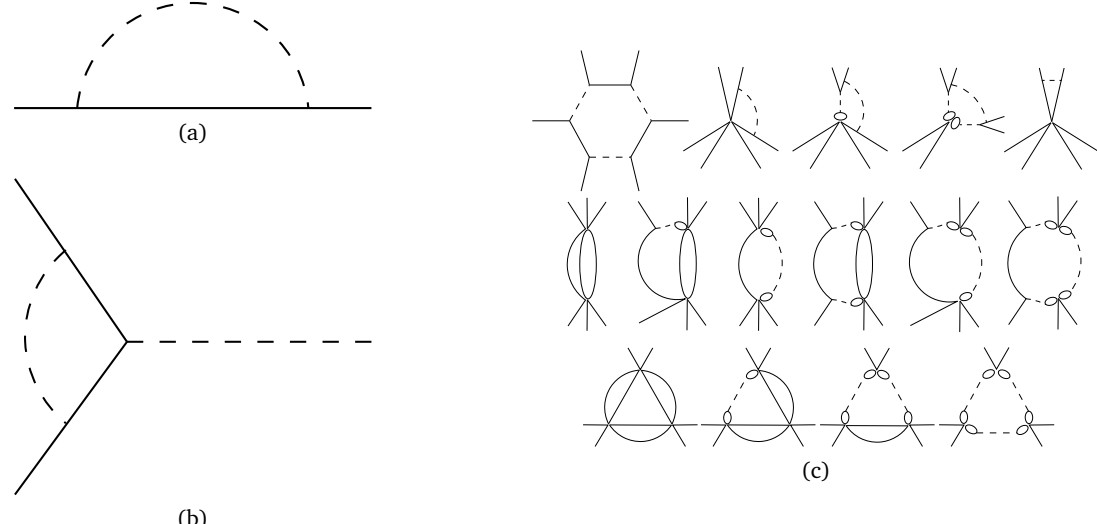

Figure 3: Diagrams that compute the renormalization constants at order $1/N$. (a) gives the wavefunction renormalization of $\varphi^I$, (b) the renormalization of the $g$ vertex, and (c) the renormalization of the $h$ vertex (permutations of external legs are omitted).

and we have a double zero at $f_g = 0$ corresponding to $g = 0$ and a simple one at $f_g = 1$ corresponding to $g = \infty$ (we do not distinguish between $+\infty$ and $-\infty$ for $g$, because of the $\mathbb{Z}_2$ symmetry the physical parameter is actually $g^2 \geq 0$ or $f_g \in [0,1]$). An RG flow connects these two points. In order to ensure that these fixed points are real, we need to check also the zeroes of the $\beta$ function of the sextic coupling, which takes the following form

$$
\begin{aligned}
\beta_h &= -\frac{d}{d \log \Lambda}(Z_\varphi^{-3} Z_h\, h) \\
&= -\frac{1}{\pi^2 N} \frac{9h^3/32 - 9h^2 - \left(g^4/2 + 4g^2 + 32\right) g^2 h + 16 g^6/3}{(1 + g^2/4)^3} + \mathcal{O}(N^{-2})\,.
\end{aligned}
\tag{2.8}
$$

Since the zeroes of $\beta_g$ are $g = 0$ and $g = \infty$, we can simply plug these values in $\beta_h$ and look for the zeroes in $h$ of the resulting cubic polynomial. For $g = 0$ it coincides with large $N$ $\beta$ function for the sextic deformation of the free 3d bosonic vector model studied in [18]. There is a UV stable fixed point at the positive value $h = 32$, and a fixed point at a double-zero for $h = 0$, which is IR stable for perturbations with positive $h$ and UV stable for perturbations with negative $h$. For $g = \infty$, the coefficients of the cubic and quadratic term in $\beta_h$ vanish and only a linear and a constant term survive. As a result there is only one IR stable fixed point for finite values of the coupling at $h = 32/3$, and in addition two UV stable fixed points at $h = \pm\infty$ (unlike $g$, for the coupling $h$ there is no symmetry flipping its sign and therefore $+\infty$ and $-\infty$ are distinct). A non-negative value of $h$ is required to ensure the stability of the vacuum.

## 2.2 Neumann coupled to $N$ critical scalars

Next, we consider the theory of a free bulk scalar with Neumann boundary condition coupled to a critical $O(N)$ model on the boundary, with the following action

$$
S_N^b[g', h'] = \int_{y \geq 0} d^3x\, dy\, \frac{1}{2}(\partial_\mu \Phi)^2 + \int_{y=0} d^3x\, \left[\frac{1}{2}\left(\partial_a \varphi^I\right)^2 + g'\sigma\Phi + \frac{1}{\sqrt{N}}\sigma\varphi^I\varphi^I\right] + S_6[h']\,.
\tag{2.9}
$$

From the point of view of $1/N$ perturbation theory $g'$ is a classically marginal coupling because the Hubbard-Stratonovich (HS) field $\sigma$ –which is the lightest $O(N)$-singlet scalar operator in the critical $O(N)$ model– has scaling dimension $2 + \mathcal{O}(N^{-1})$. The action $S_6$ contains the marginal sextic interaction and various additional classically marginal couplings that we can construct with the boundary value of the scalar field, namely

$$S_6[h'] = \int_{y=0} d^3x \left[ \frac{h'_1}{\sqrt{N}} \Phi^3 + \frac{h'_2}{N} \Phi^2 \varphi^I \varphi^I + \frac{h'_3}{N^{\frac{3}{2}}} \Phi (\varphi^I \varphi^I)^2 + \frac{h'_4}{N^2} (\varphi^I \varphi^I)^3 \right] . \qquad (2.10)$$

The scaling with $N$ is chosen so that the theory has a solvable large $N$ limit with $h'_{1,2,3,4}$ kept fixed. Using the equation of motion obtained by varying the action with respect to $\sigma$

$$g' \Phi|_{y=0} = -\frac{1}{\sqrt{N}} \varphi^I \varphi^I , \qquad (2.11)$$

we see that the four operators in $S_6$ are actually redundant, and only one coupling is physical. One can then change basis of operators to a basis with three operators that vanish on-shell, and a fourth physical operator. The physical coupling is the coefficient of the latter, and (up to rescaling by the other physical coupling $g'$) it is given by the following linear combination

$$h' = -\frac{h'_1}{g'^3} + \frac{h'_2}{g'^2} - \frac{h'_3}{g'} + h'_4 . \qquad (2.12)$$

Hence, similarly to the theory studied in the previous section, in this theory we have two classically marginal couplings $g'$ and $h'$. Also in this theory the bulk $\mathbb{Z}_2$ symmetry is broken and it acts on the couplings as $(g', h') \to (-g', h')$. The boundary interaction gives rise to a "modified Neumann" boundary condition

$$\partial_y \Phi|_{y=0} = g' \sigma + g'^2 \partial_{g'} h' \frac{1}{N^{\frac{3}{2}}} (\varphi^I \varphi^I)^2 , \qquad (2.13)$$

where $\partial_{g'} h'$ is with $h'_{1,2,3,4}$ constant.

At the leading order at large $N$ the interactions in $S_6$ can be neglected and the only effect of the $g'$ interaction is to modify the boundary two-point functions of the HS field $\sigma$ and of $\Phi$ to

$$\begin{aligned}
\langle \Phi(p) \Phi(-p) \rangle &= \frac{1}{1 + 4g'^2} \frac{1}{|p|} , \\
\langle \sigma(p) \sigma(-p) \rangle &= \frac{1}{1 + 4g'^2} (-4|p|) , \\
\langle \Phi(p) \sigma(-p) \rangle &= \frac{4g'}{1 + 4g'^2} .
\end{aligned} \qquad (2.14)$$

Therefore at the leading order $g'$ and $h'$ have vanishing $\beta$ functions and they are free parameters, similarly to $g$ and $h$ in the theory of the previous subsection.

### 2.2.1 Beta functions and anomalous dimension

We now proceed to obtain the RG functions at order $1/N$, using the large $N$ propagators (2.14). The Feynman rules are given in appendix A. It is easy to check that the couplings in $S_6$ do not contribute to the $\beta$ function for the coupling $g'$. This coupling can be seen as a quadratic mixing between $\Phi$ and $\sigma$ and there is no renormalization of the associated vertex at order

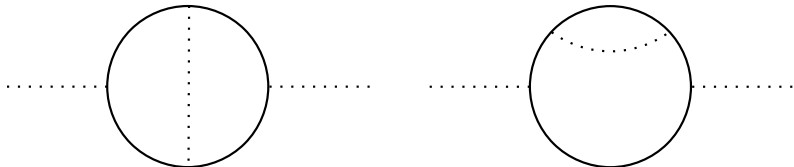

Figure 4: Diagrams that compute the renormalization of the HS field $\sigma$ at order $1/N$. This renormalization completely determines the $\beta$ function of the coupling $g'$. The dotted line denotes the propagator of $\sigma$ at the leading order at large $N$.

$1/N$. Moreover the boundary value $\Phi$ of the bulk field cannot get renormalized, so the only possible contribution is from the renormalization of the $\sigma$ operator, i.e.

$$\sigma_{\Lambda'} = Z_\sigma^{1/2}\sigma_\Lambda \,. \tag{2.15}$$

The relevant diagrams are shown in fig. 4. Similarly, the couplings in $S_6$ do not affect the wavefunction renormalization $Z_\varphi$ of the vector field at this order, which only receives contribution from the diagram analogous to that in fig. 3 (a) but with the field $\sigma$ in the internal line. The resulting renormalization constants are

$$\delta Z_\varphi = \frac{8}{3\pi^2 N}\frac{1}{1+4g'^2}\log(\Lambda/\Lambda')\,,\ \ \delta Z_\sigma = -\frac{64}{3\pi^2 N}\frac{1}{1+4g'^2}\log(\Lambda/\Lambda')\,, \tag{2.16}$$

which give the following anomalous dimension and $\beta$ function

$$\gamma_\varphi = \frac{d\log Z_\varphi^{1/2}}{d\log\Lambda} = \frac{4}{3\pi^2 N}\frac{1}{1+4g'^2}\,,$$

$$\beta_{g'} = -\frac{d}{d\log\Lambda}\left(Z_\sigma^{-1/2}g'\right) = -\frac{32}{3\pi^2 N}\frac{g'}{1+4g'^2}\,. \tag{2.17}$$

We see that the coupling $g'$ is *marginally relevant* in the vicinity of the decoupling limit $g' = 0$, and that there are two zeroes of $\beta_{g'}$, a UV stable fixed point at $g' = 0$ and an IR stable one at $g' = \infty$.

The computation of the $\beta$ function for $h'$ in principle requires the inclusion of the full set of couplings in $S_6$. In terms of the redundant basis of operators above, the relation (2.12) gives

$$\beta_{h'}(g',h') = \beta_{g'}\partial_{g'}h' - \frac{\beta_{h'_1}}{g'^3} + \frac{\beta_{h'_2}}{g'^2} - \frac{\beta_{h'_3}}{g'} + \beta_{h'_4}\,. \tag{2.18}$$

The decoupling of the operators vanishing on the equations of motion requires that the combination of $\beta$ functions on the right-hand side of eq. (2.18) must be a function of the physical couplings $g'$ and $h'$ only, as we expressed explicitly on the left-hand side. This is a non-trivial requirement, because each $\beta$ function by itself will depend on all the couplings $h'_{1,2,3,4}$, not just on their linear combination that defines $h'$. Rather than computing the full set of $\beta$ functions, we can exploit this fact to simplify our task, together with the observation that there is no diagram for $\beta_{h'_1}$, $\beta_{h'_2}$ or $\beta_{h'_3}$ at order $1/N$ which only contains the interaction vertices $h'_4$ and $g'$. Therefore setting $h'_1 = h'_2 = h'_3 = 0$ in eq. (2.18) the first four terms drop and we obtain

$$\beta_{h'}(g',h'_4) = \beta_{h'_4}|_{h'_1,h'_2,h'_3=0}\,. \tag{2.19}$$

We see that the computation of $\beta_{h'_4}$ as a function of $h'_4$ and $g'$ in the theory with $h'_1 = h'_2 = h'_3 = 0$ is enough to fix the full function $\beta_{h'}(g',h')$. The diagrams that compute the renormalization of the $h'_4$ vertex in the theory with $h'_1 = h'_2 = h'_3 = 0$ are in one-to-one correspondence with

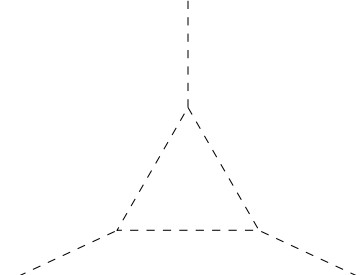

Figure 5: The only diagram that computes the $\beta$ function of the cubic self-interaction $h'_1 \Phi^3$ in the decoupling limit $g' \to 0$.

the diagrams that compute the renormalization of the $h$ vertex in the theory discussed in the previous subsection, shown in fig. 3 (c), with the difference that now the dashed propagator is interpreted as the propagator of the HS field $\sigma$. With the appropriate substitution in the formula for $\delta Z_h h$ of eq. (2.5), we then obtain

$$\delta Z_{h'_4} h'_4 \big|_{h'_{1,2,3}=0} = \frac{1}{\pi^2 N} \frac{18 g'^6 h'^3_4 - 576 g'^6 h'^2_4 - 8(240 g'^4 + 24 g'^2 + 3) h'_4 + 1024/3}{(1+4g'^2)^3} \log(\Lambda/\Lambda') ,$$

(2.20)

from which we get

$$\begin{aligned}
\beta'_h &= -\frac{d}{d \log \Lambda} \left( Z_\varphi^{-3} Z_{h'_4} h'_4 \right) \Big|_{h'_4 = h'} \\
&= \frac{1}{\pi^2 N} \frac{-18'^6 h'^3 + 576'^6 h'^2 + 32 \left( 64 g'^4 + 8 g'^2 + 1 \right) h' - 1024/3}{(1+4g'^2)^3} + \mathcal{O}(N^{-2}) .
\end{aligned}$$

(2.21)

Note that this $\beta$ function is identical to that of $h$ in the previous subsection upon the substitution $g = 1/g'$ and $h = h'$, therefore the discussion of the zeroes can be borrowed from there.

A non-trivial consistency check of this $\beta$ function comes from the decoupling limit $g' \to 0$, in which we have two separate sectors on the boundary, i.e. the 3d critical bosonic vector model and the Neumann boundary condition for the bulk free scalar. In this theory the operator $\varphi^I \varphi^I$ is set to zero by the equation of motion of $\sigma$, so among the operators in $S_6$ the only physical one is the self-interaction $\Phi^3$ of the boundary mode of the scalar field, with coupling $h'_1$. Its $\beta$ function is computed by the diagram in fig. 5, and it gives[3]

$$\beta_{h'_1} \Big|_{g'=0} = -\frac{18}{\pi^2 N} h'^3_1 + \mathcal{O}(N^{-2}) .$$

(2.22)

In passing, it is an interesting observation in its own right that for a 4d scalar field in half space with Neumann boundary condition $\Phi^3$ on the boundary is a marginally relevant deformation. From (2.18) we see that on the other hand we must have

$$\beta_{h'_1} \Big|_{g'=0} = \lim_{g' \to 0} \left[ \frac{3}{g'} \beta_{g'} h'_1 - g'^3 \beta_{h'} \left( g', -\frac{h'_1}{g'^3} \right) \right] ,$$

(2.23)

and it can be easily checked that plugging (2.21) one indeed finds the same result as the direct calculation (2.22). We conclude that the discussion of the zeroes of $\beta_{h'}$ for $g' = 0$ is more clear

---

[3]In the decoupling limit obviously $N$ is not really a parameter of the bulk free-scalar sector, and if one wishes it can be reabsorbed in the definition of the self-interaction $h'_1$. Perturbation theory in $1/N$ becomes equivalent to perturbation theory in powers of $h'^2_1$. We will keep the normalization with the inverse power of $N$ just for the ease of comparison.

Table 1: The duality map.

| $S_D^b[g,h]$ | $S_N^b[g',h']$ |
|---|---|
| $g$ | $1/g'$ |
| $h$ | $h'$ |
| $\frac{1}{\sqrt{N}}\varphi^I\varphi^I$ | $-g'\Phi\|_{y=0}$ |
| $g\,\partial_y\Phi\|_{y=0}$ | $\sigma$ |

in terms of the rescaled variable $h_1' = -\lim_{g'\to 0} g'^3 h'$, for which we have a triple zero at $h_1' = 0$, which corresponds simply to tuning to 0 the coupling of the marginally relevant operator $\Phi^3$, i.e. it is a UV stable fixed point. Note that any non-zero value of $h_1'$ would make the boundary potential unbounded from below.

## 2.3 Derivation of the duality

The two 4d/3d large $N$ theories described in the previous subsections are in fact dual descriptions of the same theory, with the dictionary between couplings and operators illustrated in table 1 and in fig. 1 in the introduction. The identification is trivial for the operators $\Phi$ and $\varphi^I$: they just map to themselves. Once we identify the operators $\Phi$ and $\varphi^I$ in the two theories, the above mapping between couplings and operators can be derived at the classical level by comparing the equations of motion and the boundary conditions. In the theory $S_D^b[g,h]$ we have

$$\Phi\big|_{y=0} = -\frac{g}{\sqrt{N}}\varphi^I\varphi^I \ , \ \ \Box\varphi^I - \frac{2g}{\sqrt{N}}\partial_y\Phi\big|_{y=0}\,\varphi^I - \frac{6h}{N^2}\left(\varphi^I\varphi^I\right)^2\varphi^I = 0 \ . \tag{2.24}$$

The first equation is the "modified Dirichlet" boundary condition while the second is the equation of motion of $\varphi^I$. In the theory $S_N^b[g',h']$ we have

$$\partial_y\Phi\big|_{y=0} = g'\sigma + g'^2\partial_{g'}h'\frac{1}{N^{\frac{3}{2}}}(\varphi^I\varphi^I)^2 \ , \ \ g'\Phi\big|_{y=0} = -\frac{1}{\sqrt{N}}\varphi^I\varphi^I \ ,$$

$$\Box\varphi^I - \frac{2}{\sqrt{N}}\sigma\varphi^I - \frac{6h' + 2g'\partial_{g'}h'}{N^2}\left(\varphi^I\varphi^I\right)^2\varphi^I = 0 \ . \tag{2.25}$$

Here again the derivative $\partial_{g'}h'$ is intended with $h'_{1,2,3,4}$ fixed. The first equation in the first line is now the "modified Neumann" boundary condition, and it allows to eliminate the additional field $\sigma$ in terms of $\partial_y\Phi$. Note that the map between $\sigma$ and $\partial_y\Phi$ can get modified at the subleading order if $h'_{1,2,3}$ are turned on. The second equation in the first line is the variation of the action with respect to $\sigma$ and it clearly maps to the boundary condition in the theory $S_D^b[g,h]$ upon the identification $g = 1/g'$. Finally comparing the equation of motion of $\varphi^I$ we obtain that $h = h'$.

We can also derive this large $N$ duality at the quantum level using a path-integral argument. Consider the partition function for the theory $S_D^b[g,h]$, in the presence of two sources $J_1, J_2$ for the operators $g\,\partial_y\Phi$ and $\frac{\varphi^I\varphi^I}{\sqrt{N}}$ respectively

$$Z_D^b[g,h,J_1,J_2] = \int_{\text{D b.c.}} [D\varphi^I][D\Phi]e^{-S_D^b[g,h] - \int_{y=0}d^3x\left[J_1 g\,\partial_y\Phi + J_2\frac{\varphi^I\varphi^I}{\sqrt{N}}\right]} \ . \tag{2.26}$$

The subscript on the right-hand side above means that the integration over $\Phi$ is restricted to those configurations that satisfy Dirichlet boundary condition at $y = 0$. Next, we integrate out the bulk fluctuations of $\Phi$, producing a non-local boundary kinetic term for $\partial_y\Phi\big|_{y=0}$. The

effective action is quadratic in $\partial_y \Phi$, so that we can perform gaussian integration over it to get (up to an irrelevant overall constant)

$$Z_D^b[g, h, J_1, J_2]$$
$$= \int [D\varphi^I][D\partial_y\Phi|_{y=0}]e^{-\int d^3x\left[\frac{1}{2}\partial_y\Phi\left(-\frac{1}{\sqrt{-\Box}}\right)\partial_y\Phi+\frac{1}{2}(\partial\varphi^I)^2+g\partial_y\Phi\frac{\varphi^I\varphi^I}{\sqrt{N}}+\frac{h}{N^2}(\varphi^I\varphi^I)^3+J_1 g\partial_y\Phi+J_2\frac{\varphi^I\varphi^I}{\sqrt{N}}\right]}$$
$$= \int [D\varphi^I]e^{-\int d^3x\left[\frac{1}{2}(\partial\varphi^I)^2+\frac{g^2}{2}\left(\frac{\varphi^I\varphi^I}{\sqrt{N}}+J_1\right)\sqrt{-\Box}\left(\frac{\varphi^J\varphi^J}{\sqrt{N}}+J_1\right)+\frac{h}{N^2}(\varphi^I\varphi^I)^3+J_2\frac{\varphi^I\varphi^I}{\sqrt{N}}\right]} .$$

$$(2.27)$$

Likewise, in the path integral of the theory $S_N^b[g', h']$, in the presence of sources $J_1'$ and $J_2'$ (this time for the operators $\sigma$ and $-g'\Phi$ respectively)

$$Z_N^b[g', h', J_1', J_2'] = \int_{\text{N b.c.}} [D\varphi^I][D\Phi]e^{-S_N^b[g',h']-\int_{y=0}d^3x\left[J_1'\sigma-J_2'g'\Phi\right]} , \qquad (2.28)$$

we integrate out the bulk components of $\Phi$, to obtain a a non-local boundary kinetic term for $\Phi|_{y=0}$. We then also integrate out the HS field $\sigma$ to get

$$Z_N^b[g', h', J_1', J_2']$$
$$= \int [D\varphi^I][D\sigma][D\Phi|_{y=0}]e^{-\int d^3x\left[\frac{1}{2}\Phi\sqrt{-\Box}\Phi+\frac{1}{2}(\partial\varphi^I)^2+g'\sigma\Phi+\sigma\frac{\varphi^I\varphi^I}{\sqrt{N}}+\frac{h'}{N^2}(\varphi^I\varphi^I)^3+J_1'\sigma-J_2'g'\Phi\right]}$$
$$= \int [D\varphi^I][D\sigma]e^{-\int d^3x\left[\frac{1}{2}(\partial\varphi^I)^2+\frac{g'^2}{2}\sigma\left(-\frac{1}{\sqrt{-\Box}}\right)\sigma+\frac{\sigma+J_2'}{\sqrt{N}}(\varphi^I\varphi^I)+\frac{h'}{N^2}(\varphi^I\varphi^I)^3+J_1'(\sigma+J_2')\right]} \qquad (2.29)$$
$$= \int [D\varphi^I]e^{-\int d^3x\left[\frac{1}{2}(\partial\varphi^I)^2+\frac{1}{2g'^2}\left(\frac{\varphi^I\varphi^I}{\sqrt{N}}+J_1'\right)\sqrt{-\Box}\left(\frac{\varphi^J\varphi^J}{\sqrt{N}}+J_1'\right)+\frac{h'}{N^2}(\varphi^I\varphi^I)^3+J_2'\frac{\varphi^I\varphi^I}{\sqrt{N}}+J_1'J_2'\right]} .$$

Here we made the additional simplifying assumption that we can take the coupling $h'$ to be associated to the sextic interaction, ignoring the three additional equations of motion-vanishing operators in the appropriate rotation of the redundant basis of operators (2.10). This can be motivated on the grounds that exchanging the order of integration over $\sigma$ and $\Phi|_{y=0}$, $\sigma$ is purely a Lagrange multiplier, so that integrating over it enforces the constraint $\Phi|_{y=0} = -\frac{g}{\sqrt{N}}\varphi^I\varphi^I$ as a functional Dirac delta. In this way the equations of motion-vanishing operators are set identically to zero when we perform the path integral over $\Phi|_{y=0}$. We see that in both theories upon some simple manipulations of the path integral we landed on the same theory of $N$ scalars with a non-local quartic interaction. Comparing (2.27) and (2.29) we obtain the following identity

$$Z_N^b[g'^2 = 1/g^2, h' = h, J_1' = J_1, J_2' = J_2] = e^{-\int d^3x J_1 J_2} Z_D^b[g^2, h, J_1, J_2] , \qquad (2.30)$$

which indeed gives the map between couplings and operators shown in table 1. We see that the there is also the shift of a contact term $\propto J_1 J_2$ when going from one description to the dual one.

Written explicitly as a convolution in position space, the non-local interaction looks like

$$\frac{g^2}{2}\int d^3x \int d^3y \frac{(\varphi^I\varphi^I)(x)}{\sqrt{N}}\left(-\frac{1}{\pi^2|x-y|^4}\right)\frac{(\varphi^J\varphi^J)(y)}{\sqrt{N}} , \qquad (2.31)$$

and we see that it has a negative-definite kernel. This could cause an instability in the theory with real $g \neq 0$ towards the generation of a condensate on the boundary, which could only be a power of the dynamically generated scale

$$M = \Lambda \exp\left[\frac{3\pi^2 N}{32}\left(\frac{2}{g^2(\Lambda)} - \log g(\Lambda)\right)\right] . \qquad (2.32)$$

The latter is non-perturbative at large $N$, i.e. the effect would not be visible at any finite order in perturbation theory. On the other hand it is not clear to us how to evaluate such a non-local potential at a constant field configuration and minimize it. Moreover, since we do not see an unbounded energy density functional in the original formulations $S_{\mathrm{D}}^{b}[g,h]$ and $S_{\mathrm{N}}^{b}[g',h']$, this negative potential might just signal a subtlety in studying the vacuum with the action obtained from "integrating out" the bulk. Further investigations are needed to clarify this issue.[4]

Some further comments about the duality are in order.

- At the leading order at large $N$ the single-trace operators behave like generalized free fields, both in the free and in the critical vector model. Therefore the interactions $g$ or $g'$ at the leading order are products of two generalized free fields with dimensions that add up to the space-time dimension, and we are in the setup of the line of fixed points with a non-perturbative duality found in [9]. To see that the duality is precisely $g' = 1/g$, it is convenient to view the setup in the same language as [9], as a theory in AdS. To this end, we use the holographic dual of the free/critical vector model, namely type A Vasiliev theory on $\mathrm{AdS}_4$ [19] (for a review, see e.g. [20]). The difference between the free/critical vector model is encoded in the choice between alternate/ordinary quantization for the scalar field in Vasiliev theory [21, 22], giving dimension 1 or 2 for the dual boundary operator. Our bulk scalar can also be placed in $\mathrm{AdS}_4$ via a Weyl rescaling of $\mathbb{R}_+ \times \mathbb{R}^3$, which endows it with a conformal mass. Similarly to the scalar in the Vasiliev sector, the difference between Neumann and Dirichlet boundary condition in flat space maps in $\mathrm{AdS}_4$ to the choice between ordinary or alternate quantization. Therefore we end up with type A Vasiliev plus a conformally coupled scalar in $\mathrm{AdS}_4$. The couplings $g$ and $g'$ can then be seen as a double-trace deformation that couples the boundary modes of the scalar in the Vasiliev sector and of the additional scalar field, taken to have opposite quantizations. At the leading order at large $N$ the scalar in Vasiliev theory is decoupled from the rest of the tower of higher spin gauge fields, therefore the two scalars are both free in the bulk and we are precisely in the AdS setup of [9]. The non-perturbative duality acts by swapping the boundary conditions for these two scalars, therefore we see that in our setup this gives $g' = 1/g$. Note that the line of fixed points is consistent with the fact that the $\beta$ functions start at the subleading order $\mathcal{O}(N^{-1})$, and the leading-order duality of [9] is consistent with the duality found above.

- A prediction of the duality is that the $g \to \infty$ limit of the theory $S_{\mathrm{D}}^{b}[g,h]$ is in fact the decoupling limit of the theory $S_{\mathrm{N}}^{b}[g',h']$. This agrees with the $g \to \infty$ limit of the leading-order two-point functions in eq. (2.3): the operators $\partial_y \Phi$ vanishes in the limit, as expected for a decoupled bulk free scalar with Neumann boundary condition, and the operator $\frac{1}{\sqrt{N}} \varphi^I \varphi^I$ also vanishes, as expected for the "naive" scalar bilinear in the critical bosonic vector model. Similar considerations apply to the $g' \to \infty$ limit of eq. (2.3), which matches with the decoupling limit of the theory $S_{\mathrm{D}}^{b}[g,h]$. These observations are all consequences of the more general fact that actually the leading order two-point functions in eq. (2.3) and eq. (2.14) match under the map. Note that the third, mixed correlator does not match. On the other hand it is a contact term so it can be shifted by a change of scheme and it does not need to match. The mismatch is

$$\langle (g \partial_y \Phi)(p) \frac{1}{\sqrt{N}} \varphi^I \varphi^I (-p) \rangle = - \langle \sigma(p)(g' \Phi)(-p) \rangle \big|_{g'=1/g} + 1 \,. \qquad (2.33)$$

We see that this shifted contact term precisely reproduces the shift $\propto J_1 J_2$ that we obtained in the path integral argument, see eq. (2.30) above.

---

[4]We thank M. Serone for discussions about this point.

- Going to the subleading order $\mathcal{O}(N^{-1})$ the line of fixed points is lifted and for both theories we have an RG. The $\beta$ functions of the couplings match under the map, see eq.s (2.6)-(2.8) and (2.17)-(2.21). For the couplings $g$ and $g'$ this may seem surprising given that the two $\beta$ functions are computed by different Feynman diagrams, compare fig. 3 (a) and (b) with fig. 4. The fact that these different diagrams give the same result can be understood as a consequence of a non-renormalization theorem for the non-local kinetic terms of $\partial_y \Phi$ or $\sigma$.[5] It is perhaps less surprising for the couplings $h$ and $h'$, given that their $\beta$ functions are computed by the same set of diagrams, namely those in fig. 3 (c). On the other hand, reducing to this set of diagrams for $h'$ required an analysis of the redundant basis of operators. Moreover, we found that $\beta_h$ (or $\beta_{h'}$) interpolates between: the $\beta$ function of $(\varphi^I \varphi^I)^3$ in the 3d theory of $N$ free scalars (at $g = 0$) and the $\beta$ function of the boundary $\Phi^3$ deformation in the theory of a 4d free scalar with Neumann boundary condition (at $g = \infty$). The duality explains the emergence of the $\beta$ function of $\Phi^3$ from the $g \to \infty$ limit of $\beta_h$, and viceversa for the theory $S_N^b[g', h']$.

- Similarly, the anomalous dimensions of the vector field $\gamma_\varphi$ match under the map, see eq.s (2.6)-(2.17). In particular, the $g \to \infty$ limit of $\gamma_\varphi$ in the theory $S_D^b[g, h]$ is finite and coincides with the order $1/N$ anomalous dimension of the vector field in the critical $O(N)$ model [23]

$$\gamma_\varphi^{O(N)} = \frac{4}{3\pi^2 N} + \mathcal{O}(N^{-2}) \,. \tag{2.34}$$

Moreover the $g \to \infty$ limit of $-\partial_g \beta_g$ gives the order $1/N$ anomalous dimension of the HS field in the critical $O(N)$ model [24, 25]

$$\gamma_\sigma^{O(N)} = -\frac{32}{3\pi^2 N} + \mathcal{O}(N^{-2}) \,. \tag{2.35}$$

The duality explains the emergence of these anomalous dimensions in the strong coupling limit of the theory $S_D^b[g, h]$.

- The hemisphere partition function was conjectured in [16] to be a quantity that decreases along boundary RG flows, and an entropic proof of this conjecture was given in [17]. We can use this monotonicity to test the existence of the boundary RG flow between the two boundary fixed points given by the decoupled theory $S_N^b[g', h']$ in the UV and the decoupled theory $S_D^b[g, h]$ in the IR. The hemisphere partition functions for the Dirichlet and Neumann boundary condition of the free scalar CFT have been computed in [16] and they are

$$F_\partial^D = -F_\partial^N = -\frac{\zeta(3)}{16\pi^2} \,. \tag{2.36}$$

On the other hand for the free bosonic vector model we have $F = N F_{\text{free scalar}}$, where for a 3d free scalar [26]

$$F_{\text{free scalar}} = \frac{\log 2}{8} - \frac{3\zeta(3)}{16\pi^2} \,, \tag{2.37}$$

and for the critical bosonic vector model the partition function up to order $\mathcal{O}(N^{-1})$ was

---

[5]The match between the logarithmic UV divergences of the diagrams in fig. 3 (a) and (b) with those of the diagrams in fig. 4 is also the reason why in the usual large $N$ perturbative approach to the critical $O(N)$ model the cubic marginal coupling $\frac{1}{\sqrt{N}} \sigma \varphi^I \varphi^I$ does not run. In that context we trade the non-renormalization of the non-local kinetic term for $\sigma$ with the non-renormalization of the cubic interaction, which can always be achieved with a rescaling, and in fact we do assign a wave-function renormalization to $\sigma$ notwithstanding its non-local kinetic term.

conjectured in [27] to be[6]

$$F_{O(N)} = N F_{\text{free scalar}} - \frac{\zeta(3)}{8\pi^2} + \frac{4}{9\pi^2 N} + \mathcal{O}(N^{-2}) \,. \tag{2.38}$$

Putting these things together we see that

$$F_{\partial}^{S_{\text{N}}^b} - F_{\partial}^{S_{\text{D}}^b} = (F_{\partial}^{\text{N}} + F_{O(N)}) - (F_{\partial}^{\text{D}} + N F_{\text{free scalar}}) = \frac{4}{9\pi^2 N} + \mathcal{O}(N^{-2}) > 0 \,. \tag{2.39}$$

The fact that the difference starts at order $1/N$ is consistent with the line of fixed points connecting the two theories at the leading order. Moreover the difference has the correct sign.

## 2.4 Adding a 4d gauge field

In this section we present a simple generalization of the theory $S_{\text{D}}^b[g,h]$ in eq. (2.1), obtained by adding the coupling to a 4d Maxwell field $A_\mu$ in the bulk. To this end, we take $N$ to be even and we organize the $N$ real scalars $\varphi^I$ into $N/2$ complex scalars as follows

$$z^m \equiv \frac{1}{\sqrt{2}} \left( \varphi^m + i \varphi^{m+N/2} \right) \,, \quad z^{\dagger m} \equiv \frac{1}{\sqrt{2}} \left( \varphi^m - i \varphi^{m+N/2} \right) \,, \quad m = 1, \ldots, N/2 \,. \tag{2.40}$$

The coupling to the 4d Maxwell field is achieved by gauging a $U(1)$ subgroup of the $O(N)$ global symmetry via the boundary component of $A_\mu$ which is taken to have Neumann boundary condition

$$F_{ya}|_{y=0} = 0 \,, \tag{2.41}$$

where $F_{\mu\nu} = \partial_\mu A_\nu - \partial_\nu A_\mu$ is the field strength. The resulting bulk/boundary action is

$$\begin{aligned} S_{\text{DN}}^b[g,h,\lambda] = \int_{y \geq 0} d^3x\, dy \left[ \frac{1}{2}(\partial_\mu \Phi)^2 + \frac{N}{4\lambda} F_{\mu\nu} F^{\mu\nu} \right] \\ + \int_{y=0} d^3x \left[ (D_a z^m)(D_a z^m)^\dagger + \frac{2g}{\sqrt{N}} \partial_y \Phi\, z^m z^{\dagger m} + \frac{8h}{N^2} (z^m z^{\dagger m})^3 \right] \,, \end{aligned} \tag{2.42}$$

where $D_a \equiv \partial_a + i A_a$ is the covariant derivative. The boundary global symmetry after the coupling to $A_\mu$ is reduced to $SU(N/2) \times U(1)$, where the $U(1)$ factor is associated to the "topological" conserved current $\propto \epsilon^{abc} F_{bc}$, and the only operators charged under it are the end-points of bulk 't Hooft lines. The coupling $\lambda$ does not run, and in the limit $\lambda \to \infty$ the bulk gauge field decouples and the boundary $U(1)$ current it couples to is gauged by emergent 3d gauge fields [3, 4]. Combining the results of the previous subsections with those of [3, 4], we conclude that at large $N$ with $\lambda \geq 0$ and $g$ held fixed, the theory (2.42) interpolates between four different bulk/boundary decoupling limits, with different 3d local CFT sectors:

- $N$ free real scalars for $g = 0$ and $\lambda = 0$ ,

- the critical $O(N)$ model for $g = \infty$ and $\lambda = 0$ ,

- critical bosonic QED$_3$ with $N/2$ flavors of complex scalars for $g = \infty$ and $\lambda = \infty$ ,

- tricritical bosonic QED$_3$ with $N/2$ flavors of complex scalars for $g = 0$ and $\lambda = \infty$ ,

as depicted in fig. 6. Here "tricritical" means that the relevant quartic interaction between the complex bosons is tuned to 0. At the leading order in the large $N$ expansion both $\lambda$ and $g$ are

---

[6]This conjecture was motivated in [27] by the analogy with certain supersymmetric models and by consistency with $\epsilon$-expansion results. However the direct calculation with large $N$ methods remained elusive, with a naive attempt producing a different result for the $1/N$ correction.

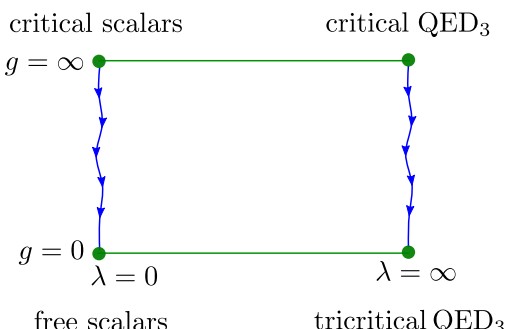

Figure 6: The four 3d CFTs connected via the interactions to the bulk scalar and the bulk gauge field. The horizontal direction represents the exactly marginal gauge coupling, while the vertical direction represents the RG flow triggered by the coupling to the bulk scalar.

marginal interactions and these four decoupling points belong to a two-parameter family of conformal boundary conditions for the free scalar plus free Maxwell fields in the bulk. The bulk fields are coupled to each other through the boundary, e.g. bulk correlation functions of local operators do not factorize in products of separate contributions from the two sectors. At the subleading order, while $\lambda$ still cannot run because it is the coefficient of a bulk operator, $g$ gets a non-trivial $\beta$ function, which will now also depend on $\lambda$, and the decoupling limits will be generically connected by RG flows. In what follows we will verify these expectations by computing the $\beta$ functions and the anomalous dimensions for a few operators of the theory (2.42), at order $\mathcal{O}(N^{-1})$. The relevant Feynman rules are given in appendix A. Note that we could have as well used the duality to the theory $S_N^b[g', h']$ before coupling to the bulk gauge field, but for definiteness we will only use the description in terms of $S_D^b[g, h]$ in this subsection.

### 2.4.1 Exact propagators at large $N$

We will need the large $N$ boundary propagators of $\partial_y \Phi$ as well as of $A_a$. The first quantity was computed earlier in this section, and the result is given in (2.3). The leading large $N$ correction to the photon propagator can be obtained in a similar way, i.e. by resumming the geometric series of the 1PI bubbles connected by tree-level photon propagators, as depicted in fig. 7. At the leading order at large $N$ the 1PI bubble of $N/2$ complex scalars is $-\frac{N|p|}{32}\left(\delta^{ab} - \frac{p^a p^b}{p^2}\right)$, so that the exact propagator of the photon at this order is (up to a longitudinal gauge-dependent term)

$$\langle A_a(p) A_b(-p) \rangle = \frac{1}{N} \frac{\lambda}{1 + \lambda/32} \frac{\delta_{ab}}{|p|}. \tag{2.43}$$

In the limit of $\lambda \to \infty$ this propagator coincides with the large $N$ effective propagator of QED$_3$, consistently with the above claim that this limit corresponds to 3d gauging.

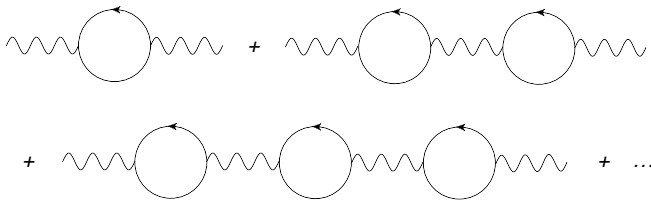

Figure 7: Diagrams that contribute to the boundary propagator of $A_a$ in the large $N$ limit with $\lambda$ fixed.

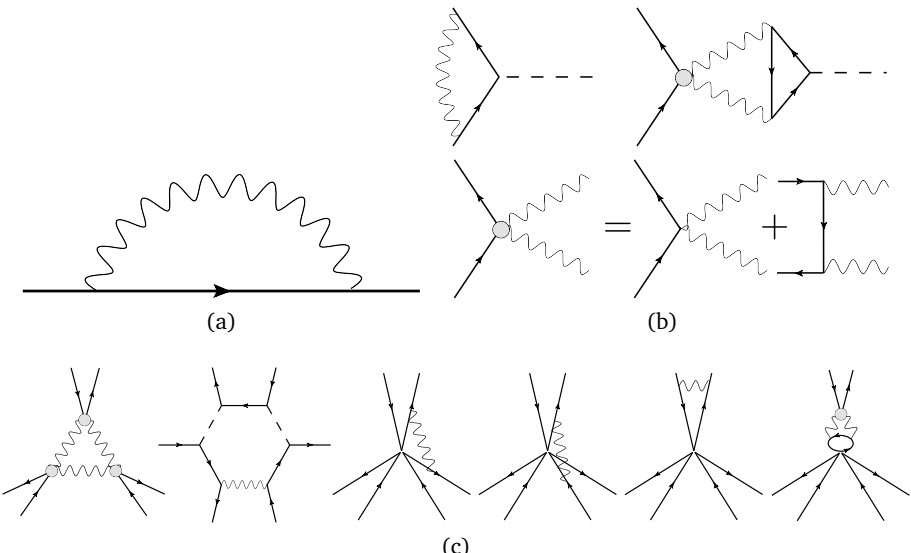

(c)

Figure 8: Contribution to the renormalization constants from the gauge field at order $1/N$. (a) gives the wavefunction renormalization of $z^m$, (b) the renormalization of the $g$ vertex, and (c) the renormalization of the $h$ vertex (permutations of external legs are omitted).

### 2.4.2 Beta functions and anomalous dimensions

Compared to the previous case without the gauge field, we have additional contributions from the diagrams depicted in fig. 8. The renormalization constants are found to be

$$\delta Z_z = \delta Z_\varphi|_{\lambda=0} - \frac{5}{6\pi^2 N}\frac{\lambda}{1+\lambda/32}\log(\Lambda/\Lambda'),$$

$$\delta Z_g g = \delta Z_g g|_{\lambda=0} + \frac{1}{2\pi^2 N}\frac{\lambda(1-7\lambda/32)g}{(1+\lambda/32)^2}\log(\Lambda/\Lambda'),$$

$$\delta Z_h h = \delta Z_h h|_{\lambda=0}$$
$$+ \frac{1}{\pi^2 N}\left(\frac{9h\lambda(1-3\lambda/16-7\lambda^2/1024)+\lambda^3}{6(1+\lambda/32)^3} - \frac{g^4\lambda}{(1+g^2/4)^2(1+\lambda/32)}\right)\log(\Lambda/\Lambda'),$$

$$(2.44)$$

where the values for $\lambda=0$ are in eq. (2.5). From these expressions we obtain the $\beta$ functions[7]

$$\beta_g = -\frac{d}{d\log\Lambda}(Z_z^{-1}Z_g g) = \beta_g|_{\lambda=0} - \frac{4}{3\pi^2 N}\frac{g\,\lambda(1-\lambda/16)}{(1+\lambda/32)^2} + \mathcal{O}(N^{-2}),$$

$$\beta_h = -\frac{d}{d\log\Lambda}(Z_z^{-3}Z_h h) = \beta_h|_{\lambda=0}$$
$$+ \frac{1}{\pi^2 N}\left(-\frac{4h\lambda(1-\lambda/16)(1+\lambda/32)+\lambda^3/6}{(1+\lambda/32)^3} + \frac{g^4\lambda}{(1+g^2/4)^2(1+\lambda/32)}\right) + \mathcal{O}(N^{-2}),$$

$$(2.45)$$

---

[7] $z^m$ is not a local gauge-invariant operator after coupling to the bulk gauge field, so its anomalous dimension is not an observable and the renormalization constants in (2.44) depend on our choice of gauge. The gauge dependence cancels in the $\beta$ functions of $g$ and $h$. One could construct a gauge-invariant non-local operator by attaching a bulk Wilson line to the insertion of $z^m$. Similarly the end-points of bulk 't Hooft lines can be seen as magnetically-charged point-like operators on the boundary, and they give the local monopole operators of the 3d gauge theory in the decoupling limit at $\lambda \to \infty$.

where the expressions for $\lambda = 0$ are given in eq.s (2.6)-(2.8). It is convenient to express $\beta_g$ in terms of the quantities $f_g = \frac{g^2/4}{1+g^2/4} \in [0,1]$ and $f_\lambda = \frac{\lambda/32}{1+\lambda/32} \in [0,1]$ as follows

$$\beta_{f_g} = \frac{64}{3\pi^2 N} f_g (1-f_g)[f_g - 4f_\lambda (1-3f_\lambda)] . \tag{2.46}$$

This formula makes more explicit that there are three distinct families of curves of fixed points in the $(f_g, h)$ plane, parametrized by $f_\lambda$

- A family with $f_g = 0$ and three possible solutions for $h = h(f_\lambda)$ from the cubic equation $\beta_h = 0$. The solutions with real $h$ are depicted in blue in the upper panel of fig. 9. These solutions correspond to unitary conformal boundary conditions for $A_\mu$ with a decoupled bulk scalar $\Phi$ with Dirichlet boundary condition.

- A family with $f_g = 1$, for which $\beta_h$ becomes linear and therefore has a single zero at $h = \frac{16(2-3f_\lambda + 32f_\lambda^3)}{3-12f_\lambda + 36f_\lambda^2}$. These solutions are depicted in orange the lower panel of fig. 9. They correspond to unitary conformal boundary conditions for $A_\mu$ with a decoupled $\Phi$ with Neumann boundary condition.

- A family with $f_g = 4f_\lambda (1-3f_\lambda)$ and three possible solutions for $h = h(f_\lambda)$ from the cubic equation $\beta_h = 0$. The solutions with real $h$ are depicted in red in the upper panel of fig. 9. This family corresponds to unitary conformal boundary conditions in which $A_\mu$ and $\Phi$ are coupled to each other through the boundary.

In the upper panel of fig. 9 we see two lines of fixed points that annihilate as we dial the exactly marginal parameter and become complex (though the annihilation happens for $g^2 < 0$). Moreover, in both panels we see instances of a couple of real solutions for the exactly marginal parameter $f_\lambda$ that annihilate and become complex as we dial the sextic coupling $h$. Therefore this simple setup might be a useful playground for the study of complex CFTs, advocated in [28, 29] to model the physics of weakly first-order transitions. Note that in this example there is no need to treat a discrete parameter as continuous, because the annihilation happens due to the presence of the exactly marginal gauge coupling (we stress that the gauge coupling is exactly marginal to all orders in perturbation theory due to locality, not as a consequence of the large $N$ limit; the role of the large $N$ limit is to allow us to explore finite values of $g$ and $h$).

The RG flows between fixed points with decoupled bulk gauge fields, i.e. with either $f_\lambda = 0$ or $f_\lambda = 1$, are depicted in fig. 10. We observe that there is no fixed point with real couplings, $f_\lambda = 0$ or $f_\lambda = 1$ and $f_g \neq 0, 1$. If such a fixed point existed, it would provide an example of a unitary, interacting (because $f_g \neq 0, 1$) and local (because the bulk gauge field is decoupled for $f_\lambda = 0$ or $f_\lambda = 1$) boundary condition for the free scalar field, of the sort that were searched recently in [15] using conformal bootstrap techniques. In the appendix B we study the generalization of the $\beta$ functions and of the fixed points to the case with a bulk $\theta$ term, and in that case we do find examples of (parity-breaking) interacting boundary conditions for the free scalar. A detailed study of these boundary conditions is left for future work.

As an interesting special case, we consider $\beta_h$ for $f_\lambda = 1$ and $f_g = 0$, in which case the bulk is completely decoupled

$$\beta_h|_{\lambda=\infty, g=0} = \frac{1}{\pi^2 N} \left( -\frac{9}{32} h^3 + 9h^2 + 256h - \frac{16384}{3} \right) + \mathcal{O}(N^{-2}) . \tag{2.47}$$

This is the $\beta$ function for the sextic coupling in tricritical bosonic QED$_3$ at large $N$, which to our knowledge has not been computed before. There are two zeroes at real positive values of

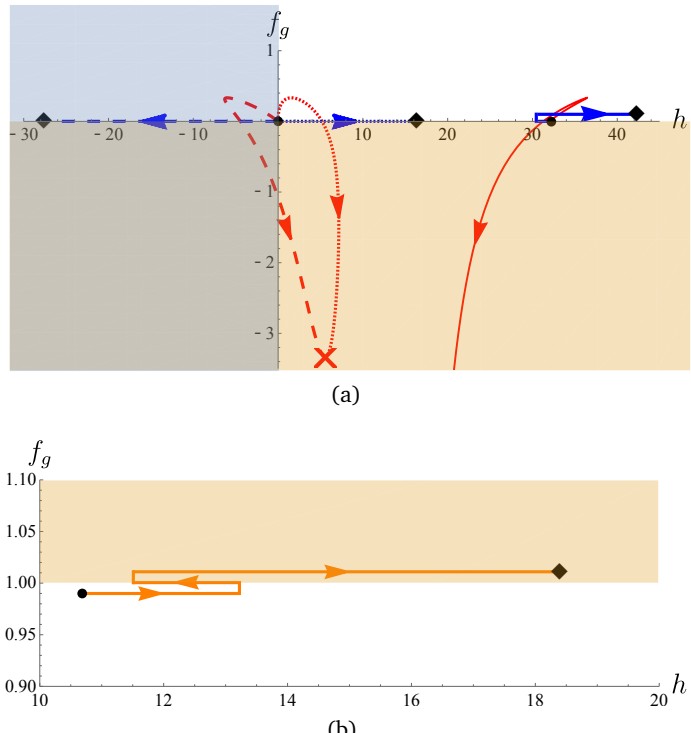

(a)

(b)

Figure 9: Families of unitary fixed points as a function of the exactly marginal parameter $f_\lambda$: for $f_g = 0$ (blue curves) and $f_g = 4f_\lambda(1-3f_\lambda)$ (red curves) in (a), for $f_g = 1$ (orange curve) in (b). The arrows denote directions of increasing $f_\lambda \in [0,1]$. The shaded blue region is $h < 0$ for which the scalar potential is not bounded from below. The shaded orange region is $f_g \notin [0,1]$ for which $g^2 < 0$. The black dots denote the fixed points (at finite values of $h$) for $f_\lambda = 0$, namely $(f_g, h) \in \{(0,0),(0,32),(1,\frac{32}{3})\}$. The black squares denote the fixed points (at finite values of $h$) for $f_\lambda = 1$, namely $(f_g, h) \in \{(0,-\frac{16}{3}(1 \pm \sqrt{17})),(0,\frac{128}{3}),(1,\frac{496}{27})\}$. For aesthetic reasons we did not show the $f_\lambda = 1$ endpoint on the solid, red curve at $(f_g, h) \approx (-8, 18.45)$. Different dashing of the lines in (a) denote the three different solutions for the cubic equation $\beta_h = 0$ for the given solution for $f_g$. The red cross at $(f_g, h) \approx (-3.33, 5.54)$ denotes the point in which two distinct zeroes of $\beta_h$ collide and the solution for $h$ becomes complex. For the solution represented by the solid blue curve, the same value of $h$ can be obtained for two distinct values of $f_\lambda$, and we added a small displacement in the vertical direction to visualize these two solutions. In the case of (b) $\beta_h$ collapses to a linear function of $h$ so there is only one solution (at finite values of $h$), however the same value of $h$ can be obtained for three distinct values of $f_\lambda$, and we added a small displacement in the vertical direction to visualize these three solutions.

$h$, one UV stable at $h_{\text{UV}} = \frac{128}{3}$ and the other IR stable at $h_{\text{IR}} = \frac{16(\sqrt{17}-1)}{3} \approx 16.66$, while $h = 0$ is not a fixed point. If we define tricritical bosonic QED$_3$ as the IR fixed point of the theory with $N/2$ complex scalars coupled to a 3d abelian gauge field with Maxwell action, with the quartic interaction fine-tuned to zero but with no further fine-tuning of the sextic, then at large $N$ this should correspond to the fixed point at $h_{\text{IR}}$.

As a check of our result, we note that in the limit (2.23) (recall that $g' = 1/g$ and $h' = h$) we obtain the $\beta$ function (2.22) of the cubic deformation $\Phi^3$ of the Neumann boundary condition, for any value of $\lambda$.

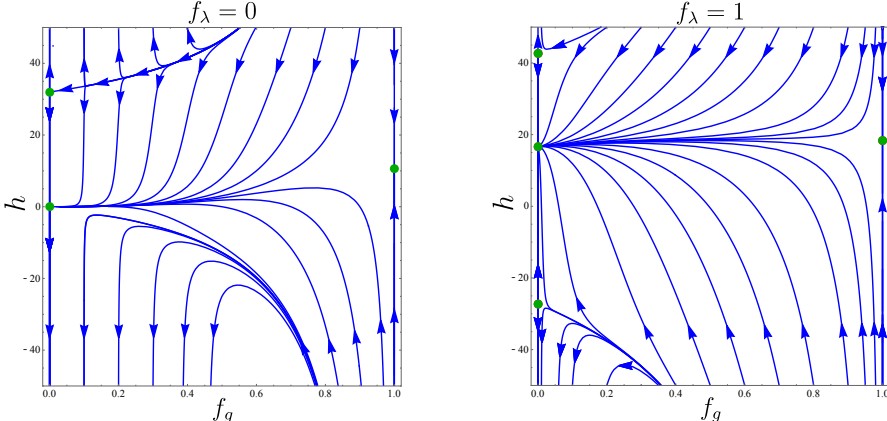

Figure 10: Boundary RG flow for $f_\lambda = 0, 1$, i.e. with the bulk gauge field decoupled. The green dots denote the fixed points (at finite values of $h$), namely $(f_g, h) \in \{(0,0), (0,32), (1,\frac{32}{3})\}$ for $f_\lambda = 0$, and $(f_g, h) \in \{(0, -\frac{16}{3}(1 \pm \sqrt{17})), (0, \frac{128}{3}), (1, \frac{496}{27})\}$ for $f_\lambda = 1$.

### 2.4.3 Anomalous dimension of lowest singlet

Let us now consider the anomalous dimension of the simplest gauge-invariant operator for the theory $S^b_{\text{DN}}[g, h, \lambda]$. Given the modified Dirichlet boundary condition

$$\Phi|_{y=0} + \frac{2g}{\sqrt{N}} z^{\dagger m} z^m = 0 , \tag{2.48}$$

and the fact that $\Phi|_{y=0}$ does not renormalize, we see that the renormalization of the operator $z^{\dagger m} z^m$ is fixed by that of the coupling, namely upon defining

$$(z^{\dagger m} z^m)_{\Lambda'} = Z_{z^\dagger z} (z^{\dagger m} z^m)_\Lambda , \tag{2.49}$$

and recalling that $g_{\Lambda'} = Z_z^{-1} Z_g g_\Lambda$ we obtain

$$Z_{z^\dagger z} = Z_z Z_g^{-1} , \tag{2.50}$$

and therefore

$$\gamma_{z^\dagger z} = \frac{d \log Z_{z^\dagger z}}{d \log \Lambda} = \frac{\beta_g}{g} . \tag{2.51}$$

This relation holds to all orders in the $1/N$ expansion because the boundary conditions and the non-renormalization of $\Phi|_{y=0}$ hold to all orders. In the appendix C we check this result in the case $\lambda = 0$ by rederiving it in dimensional regularization. Plugging the result in eq. (2.45) for $\beta_g$ up to order $\mathcal{O}(N^{-1})$ we obtain

$$\gamma_{z^\dagger z} = \frac{1}{12\pi^2 N} \left( \frac{32g^2}{1 + g^2/4} - \frac{16\lambda(1 - \lambda/16)}{(1 + \lambda/32)^2} \right) + \mathcal{O}(N^{-2}) . \tag{2.52}$$

This function determines an observable scaling dimension when evaluated at the fixed points. Note that both for the family of fixed points at $g = 0$ and for the family at $g = \infty$ it coincides with the partial derivative $\partial_g \beta_g$ evaluated at those fixed points, while $\gamma_{z^\dagger z}$ vanishes identically for the family of fixed point corresponding to the non-trivial zero $g = g(\lambda) \neq 0, \infty$. In the $\lambda \to \infty$ limit of the fixed points at $g = 0$ we find

$$\gamma_{z^\dagger z}|_{g=0, \lambda=\infty} = \frac{256}{3\pi^2 N} + \mathcal{O}(N^{-2}) . \tag{2.53}$$

This result matches the large $N$ anomalous dimension of the mass operator in the tricritical bosonic QED$_3$ with $N/2$ complex scalars [30].

In the family of fixed points with $g = \infty$ the operator $z^\dagger z$ vanishes, which can be seen either from the fact that the normalization of its two-point function approaches zero in the limit $g \to \infty$, or from the perspective of the dual theory from the equation of motion of the HS field. On the other hand we can still read off the dimension of the lowest lying singlet scalar operator from the limit of the function $\gamma_{z^\dagger z}$, only now it gives the opposite of the anomalous dimension. Again this can be derived in two ways: either noticing that this operator is furnished by $g \partial_y \Phi|_{y=0}$, which in the limit $g \to \infty$ decouples from the bulk scalar field, and then using the fact that $\partial_y \Phi|_{y=0}$ does not renormalize to relate the anomalous dimension of this operator to $\beta_g$; or using the perspective of the dual theory, in which this operator is the HS field $\sigma$, and then exploiting that $\gamma_\sigma = -\gamma_{z^\dagger z}$.[8] Indeed we have

$$-\gamma_{z^\dagger z}|_{g=\infty,\lambda=0} = -\frac{32}{3\pi^2 N} + \mathcal{O}(N^{-2}) \, , \tag{2.54}$$

matching the scaling dimension of the HS field in the critical $O(N)$ model [24, 25], and

$$-\gamma_{z^\dagger z}|_{g=\infty,\lambda=\infty} = -\frac{96}{\pi^2 N} + \mathcal{O}(N^{-2}) \, , \tag{2.55}$$

matching the scaling dimension of the HS field in the critical bosonic QED$_3$ with $N/2$ complex scalars [31]. We observe that since at this order $\gamma_{z^\dagger z}$ is the the sum of two separate functions of $g$ and $\lambda$ that vanish when $g = 0$ or $\lambda = 0$ we have

$$\gamma_{z^\dagger z}|_{g=\infty,\lambda=\infty} = \gamma_{z^\dagger z}|_{g=\infty,\lambda=0} + \gamma_{z^\dagger z}|_{g=0,\lambda=\infty} \, . \tag{2.56}$$

This explains the additive relation between the $\mathcal{O}(N^{-1})$ anomalous dimensions in the critical $O(N)$ model, tricritical bosonic QED$_3$ and critical bosonic QED$_3$. It would be interesting to see if this structure persists at higher orders in the $1/N$ expansion.

# 3 Large $N$ fermions on the boundary

In this section we consider the analogous construction in the case of the fermionic vector model. The story is very similar to the bosonic case, and even a bit simpler because there is no analogue of the sextic interaction due to an additional $\mathbb{Z}_2$ symmetry, so even though the presentation will be self-contained and with the same organization as in the previous section, we will keep the details to a minimum.

## 3.1 Neumann coupled to $N$ free fermions

We start by considering a 4d bulk scalar field $\Phi$ with Neumann boundary condition, coupled to $N$ 3d free Majorana fermions $\psi^I$ on the boundary. The bulk/boundary action is

$$S^f_{\mathrm{N}}[g] = \int_{y\geq 0} d^3 x \, dy \, \frac{1}{2}(\partial_\mu \Phi)^2 + \int_{y=0} d^3 x \left[ \bar{\psi}^I \slashed{\partial} \psi^I + \frac{g}{\sqrt{N}} \Phi \, \bar{\psi}^I \psi^I \right] . \tag{3.1}$$

Besides the $O(N)$ global symmetry that rotates the $\psi^I$'s, we also have an unbroken diagonal $\mathbb{Z}_2$ that combines reflections on the boundary, under which $\bar{\psi}^I \psi^I$ is odd, and the sign flip of the scalar $\Phi \to -\Phi$. The operator $\Phi^3$ cannot be generated due to this $\mathbb{Z}_2$ symmetry and therefore the action above contains all the possible marginal interactions. We see here an important difference compared to the bosonic case, namely there is only one coupling. The Feynman rules are given in appendix A.

---

[8]This can be derived for instance observing that $\langle \sigma(x)(z^\dagger z)(y) \rangle \propto \delta^3(x-y)$ so the only sensible assignment of scaling dimension to the vanishing operator $z^\dagger z$ is $3 - \Delta_\sigma$.

### 3.1.1 Exact two-point functions, beta function and anomalous dimension

The bubble diagram with one Majorana fermion gives $-|p|/16$, and the boundary propagator of $\Phi$ for a scalar with Neumann boundary condition is $1/|p|$, therefore resumming the bubble diagrams depicted in fig. 11 we find

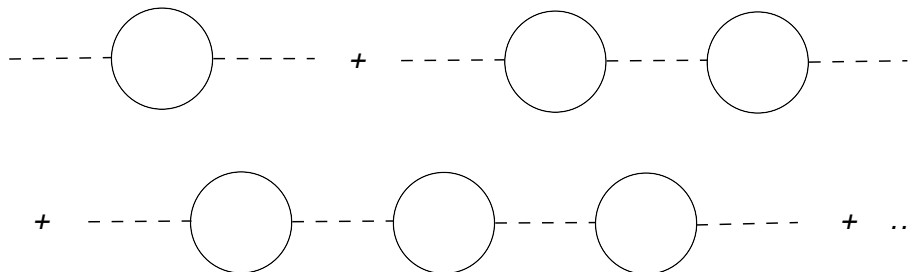

Figure 11: Diagrams that contribute to the boundary propagator of $\Phi$ in the limit of large $N$ with $g$ fixed.

$$\langle \Phi(p)\Phi(-p)\rangle = \frac{1}{1+g^2/16}\frac{1}{|p|} \,,$$

$$\langle \frac{1}{\sqrt{N}}\bar{\psi}^I\psi^I(p)\frac{1}{\sqrt{N}}\bar{\psi}^J\psi^J(-p)\rangle = \frac{1}{1+g^2/16}(-|p|/16) \,, \qquad (3.2)$$

$$\langle \Phi(p)\frac{1}{\sqrt{N}}\bar{\psi}^I\psi^I(-p)\rangle = \frac{g/16}{1+g^2/16} \,.$$

While at the leading order $g$ parametrizes a line of fixed points, at order $\mathcal{O}(N^{-1})$ most of these fixed points are lifted by the $\beta$ function. The diagrams that contribute to the anomalous dimension of $\psi^I$ and to the $\beta$ function at order $\mathcal{O}(N^{-1})$ are depicted in fig. 12. Evaluating

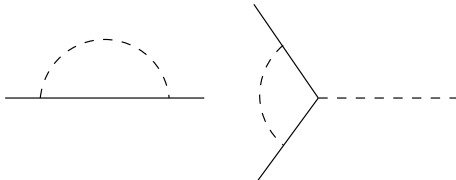

Figure 12: Diagrams that compute the renormalization constants at order $1/N$.

them we find

$$\gamma_\psi = \frac{1}{12\pi^2 N}\frac{g^2}{1+g^2/16} + \mathcal{O}(N^{-2}) \,,$$

$$\beta_g = \frac{2}{3\pi^2 N}\frac{g^3}{1+g^2/16} + \mathcal{O}(N^{-2}) \,. \qquad (3.3)$$

Therefore there is an IR stable fixed point at $g = 0$ and a UV stable one at $g = \infty$. This is more transparent in the "compactified" variable $f_g = \frac{g^2/16}{1+g^2/16} \in [0,1]$ in terms of which we have

$$\beta_{f_g} = \frac{64}{3\pi^2 N}f_g^2(1-f_g) + \mathcal{O}(N^{-2}) \,. \qquad (3.4)$$

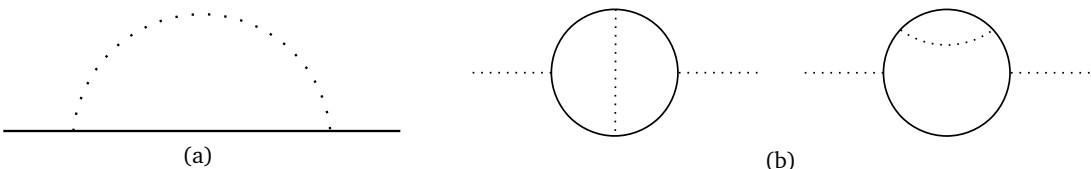

Figure 13: Diagrams that compute the renormalization constants at order $1/N$. (a) gives the wavefunction renormalization of $\psi^I$, (b) the wavefunction renormalization of $\sigma$.

## 3.2 Dirichlet coupled to $N$ critical fermions

Next, we consider a 4d bulk scalar with Dirichlet boundary condition, coupled to the 3d $O(N)$-symmetric Gross-Neveu model on the boundary

$$S_{\text{D}}^f[g'] = \int_{y\geq 0} d^3x\,dy\,\frac{1}{2}(\partial_\mu\Phi)^2 + \int_{y=0} d^3x\left[\bar{\psi}^I\!\!\not{\partial}\psi^I + g'\,\sigma\,\partial_y\Phi + \frac{1}{\sqrt{N}}\,\sigma\,\bar{\psi}^I\psi^I\right]. \quad (3.5)$$

The HS field $\sigma$ has scaling dimension $1 + \mathcal{O}(N^{-1})$ at large $N$, so that $g'$ is classically marginal. As in the previous subsection, there is an unbroken diagonal $\mathbb{Z}_2$ between the reflection symmetry on the boundary, under which $\sigma$ is odd, and the sign flip of the bulk scalar. Therefore the operator $\sigma^3$ cannot be generated, and there is only one marginal coupling $g'$. The Feynman rules are given in appendix A.

### 3.2.1 Exact two-point functions, beta function and anomalous dimension

At the leading order in the large $N$ limit the only effect of the $g'$ interaction is to modify the two-point functions with $\sigma$ and $\partial_y\Phi$

$$\langle\partial_y\Phi(p)\partial_y\Phi(-p)\rangle = \frac{1}{1+16g'^2}(-|p|)\,,$$

$$\langle\sigma(p)\sigma(-p)\rangle = \frac{16}{1+16g'^2}\frac{1}{|p|}\,, \quad (3.6)$$

$$\langle\partial_y\Phi(p)\sigma(-p)\rangle = \frac{16g'}{1+16g'^2}\,.$$

The coupling $g'$ parametrizes a line of fixed points at the leading order at large $N$. To compute $\beta_{g'}$ and $\gamma_\psi$ at order $\mathcal{O}(N^{-1})$, we use the large $N$ propagators (3.6). Like in bosonic model of section 2.2, the interaction $g'$ is a quadratic mixing between $\partial_y\Phi$ and $\sigma$, and its renormalization comes only from the wavefunction renormalization of $\sigma$. Evaluating the diagrams in fig. 13 we obtain

$$\gamma_\psi = \frac{4}{3\pi^2 N}\frac{1}{1+16g'^2}\,,$$

$$\beta_{g'} = -\frac{32}{3\pi^2 N}\frac{g'}{1+16g'^2}\,. \quad (3.7)$$

There is an IR stable fixed point at $g' = 0$ and a UV stable one at $g' = \infty$.

## 3.3 Derivation of the duality

The two large $N$ theories $S_N^f[g]$ and $S_D^f[g']$ are dual to each other. The dictionary between the couplings and the operators is summarized in table 2. Note that $\Phi$ and $\psi^I$ map to themselves

Table 2: The duality map between the two large $N$ fermionic theories.

| $S_{\mathrm{N}}^{f}[g]$ | $S_{\mathrm{D}}^{f}[g']$ |
|---|---|
| $g$ | $-1/g'$ |
| $\frac{1}{\sqrt{N}}\bar{\psi}^{I}\psi^{I}$ | $-g'\,\partial_{y}\Phi\|_{y=0}$ |
| $g\,\Phi\|_{y=0}$ | $\sigma$ |

under the duality. The dictionary can be derived at the classical level by comparing the equations of motion and the boundary conditions. On the Neumann side, when we vary the action $S_{\mathrm{N}}^{f}[g]$ with respect to the bulk scalar and the fermions we obtain the "modified Neumann" boundary condition as well as the equation of motion for $\psi^{I}$

$$\partial_{y}\Phi\big|_{y=0} - \frac{g}{\sqrt{N}}\bar{\psi}^{I}\psi^{I} = 0\,, \quad \slashed{\partial}\psi^{I} + \frac{g}{\sqrt{N}}\Phi\big|_{y=0}\,\psi^{I} = 0\,. \tag{3.8}$$

On the Dirichlet side, from varying the action $S_{\mathrm{D}}^{f}[g']$ we find the "modified Dirichlet" boundary condition, as well as the equation of motion for $\psi^{I}$ and $\sigma$

$$\Phi\big|_{y=0} + g'\sigma = 0\,, \quad \slashed{\partial}\psi^{I} + \frac{1}{\sqrt{N}}\sigma\,\psi^{I} = 0\,, \quad g'\partial_{y}\Phi\big|_{y=0} + \frac{1}{\sqrt{N}}\bar{\psi}^{I}\psi^{I} = 0\,. \tag{3.9}$$

We can solve the leftmost equation above to eliminate $\sigma$. The resulting system is now completely equivalent to what we obtained for the theory $S_{\mathrm{N}}^{f}[g]$ upon identifying $g = -1/g'$.[9]

At the quantum level, the duality follows from simple path integral manipulations, in complete analogy to what we have done for scalars (see section 2.3). In particular, starting from the partition function for the theory $S_{\mathrm{N}}^{f}[g]$ in the presence of two sources $J_{1}, J_{2}$ for the operators $g\Phi$ and $\frac{\bar{\psi}^{I}\psi^{I}}{\sqrt{N}}$, respectively, and integrating out $\Phi$ we obtain

$$\begin{aligned} Z_{\mathrm{N}}^{f}[g, J_{1}, J_{2}] &= \int_{\mathrm{N\,b.c.}} [D\psi^{I}][D\Phi]e^{-S_{\mathrm{N}}^{f}[g] - \int_{y=0} d^{3}x \left[J_{1}\,g\Phi + J_{2}\frac{\bar{\psi}^{I}\psi^{I}}{\sqrt{N}}\right]} \\ &= \int [D\psi^{I}]e^{-\int d^{3}x \left[\bar{\psi}^{I}\slashed{\partial}\psi^{I} - \frac{g^{2}}{2}\left(\frac{\bar{\psi}^{I}\psi^{I}}{\sqrt{N}} + J_{1}\right)\frac{1}{\sqrt{-\Box}}\left(\frac{\bar{\psi}^{J}\psi^{J}}{\sqrt{N}} + J_{1}\right) + J_{2}\frac{\bar{\psi}^{I}\psi^{I}}{\sqrt{N}}\right]}\,. \end{aligned} \tag{3.10}$$

Similarly for the path integral of the theory $S_{\mathrm{D}}^{f}[g']$ in the presence of sources $J_{1}'$ and $J_{2}'$ (this time for the operators $\sigma$ and $-g'\partial_{y}\Phi$ respectively), integrating out $\Phi$ and then $\sigma$ we obtain

$$\begin{aligned} Z_{\mathrm{D}}^{f}[g', J_{1}', J_{2}'] &= \int_{\mathrm{D\,b.c.}} [D\psi^{I}][D\Phi]e^{-S_{\mathrm{D}}^{f}[g] - \int_{y=0} d^{3}x \left[J_{1}'\sigma - J_{2}'\,g'\partial_{y}\Phi\right]} \\ &= \int [D\psi^{I}]e^{-\int d^{3}x \left[\bar{\psi}^{I}\slashed{\partial}\psi^{I} - \frac{1}{2g'^{2}}\left(\frac{\bar{\psi}^{I}\psi^{I}}{\sqrt{N}} + J_{1}'\right)\frac{1}{\sqrt{-\Box}}\left(\frac{\bar{\psi}^{J}\psi^{J}}{\sqrt{N}} + J_{1}'\right) + J_{2}'\frac{\bar{\psi}^{I}\psi^{I}}{\sqrt{N}} + J_{1}'J_{2}'\right]}\,. \end{aligned} \tag{3.11}$$

Therefore we have

$$Z_{\mathrm{D}}^{f}[g'^{2} = 1/g^{2}, J_{1}' = J_{1}, J_{2}' = J_{2}] = e^{-\int d^{3}x\, J_{1}J_{2}}\, Z_{\mathrm{N}}^{f}[g^{2}, J_{1}, J_{2}]\,. \tag{3.12}$$

Also in this theory the non-local kernel in the quartic interaction is negative, and one might worry that a non-perturbative condensate is generated for the fermionic bilinear. As in the

---

[9]The relative minus sign compared to the bosonic theory is due to fact that we are using always positive signs in the actions to define our couplings, and that while the deformation of a Neumann boundary condition by the boundary interaction $\Phi O_{\mathrm{N}}$ leads to the modified Neumann boundary condition $\partial_{y}\Phi\|_{y=0} = O_{\mathrm{N}}$, the deformation of a Dirichlet boundary condition by the boundary interaction $\partial_{y}\Phi O_{\mathrm{D}}$ leads to the modified Dirichlet boundary condition $\Phi\|_{y=0} = -O_{\mathrm{D}}$, with an additional sign compared to the Neumann case.

bosonic case however we do not see signs of such an instability in the original formulations $S_\text{N}^f[g]$ and $S_\text{D}^f[g']$ and therefore suspect this is due to the procedure of "integrating out" the bulk.

Most of the comments we made about the duality in section 2.3 apply to this case as well, modulo the appropriate changes. Let us briefly mention them and refer to that section for a more detailed discussion.

- The line of fixed points and the duality $g = -1/g'$ at the leading order at large $N$ can again be seen as a special case of the setup discussed in [9]. Also in this case we can view the fermionic vector model in terms of its bulk dual, with the difference that now the dual is type B Vasiliev theory [32, 33], and place the bulk scalar in $AdS_4$ via a Weyl rescaling. It is still true that at the leading order we are just left with two conformally coupled bulk scalars, with opposite choice for the boundary conditions, coupled to each other through a mixed boundary condition.

- The leading large $N$ two-point functions at separated points in eq. (3.6) and eq. (3.2) match under the map of table 2. As in the scalar case, the contact terms in the third lines of (3.6)-(3.2) do not map to each other. This shift of these contact terms under the duality precisely agrees with the one $\propto J_1 J_2$ from the path integral argument, see eq. (3.12).

- The $\beta$ functions of the two theories, as well as the the anomalous dimensions of the fermions $\gamma_\psi$ match under the map, see eq.s (3.3)-(3.7). In particular, the $g \to \infty$ limits of $\gamma_\psi$ and $\partial_g \beta_g$ in the theory $S_\text{N}^f[g]$ reproduce, respectively, the $\mathcal{O}(N^{-1})$ anomalous dimensions of the fermions and of the HS field in the $O(N)$ Gross-Neveu model [34, 35]

$$\gamma_\psi^\text{GN} = \frac{4}{3\pi^2 N} + \mathcal{O}(N^{-2}) \, , \quad \gamma_\sigma^\text{GN} = -\frac{32}{3\pi^2 N} + \mathcal{O}(N^{-2}) \, . \tag{3.13}$$

- Like in the bosonic theory, we can test the existence of the RG between the decoupled theory $S_\text{D}^f[g' = 0]$ in the UV and the decoupled theory $S_\text{N}^f[g = 0]$ in the IR using the boundary F-theorem [16,17]. For for the free fermionic vector model we have $F = N F_\text{free fermion}$, where [26]

$$F_\text{free fermion} = \frac{\log 2}{8} + \frac{3\zeta(3)}{16\pi^2} \, , \tag{3.14}$$

and for the critical fermionic vector model we have [27]

$$F_\text{GN} = N F_\text{free fermion} + \frac{\zeta(3)}{8\pi^2} + \frac{4}{9\pi^2 N} + \mathcal{O}(N^{-2}) \, . \tag{3.15}$$

Recalling also (2.36), we obtain

$$F_\partial^{S_\text{D}^f} - F_\partial^{S_\text{N}^f} = (F_\partial^\text{D} + F_\text{GN}) - (F_\partial^\text{N} + N F_\text{free fermion}) = \frac{4}{9\pi^2 N} + \mathcal{O}(N^{-2}) > 0 \, , \tag{3.16}$$

which has the correct sign to allow the RG.

## 3.4 Adding a 4d gauge field

We now consider the coupling of the theory $S_\text{N}^f[g]$ to a 4d Maxwell field $A_\mu$ with Neumann boundary condition. This coupling comes from gauging a $U(1)$ subgroup of the $O(N)$ global symmetry using the boundary limit of the bulk gauge field. It is convenient to reorganize the fermions in $N/2$ Dirac fields ($N/2$ is taken to be even to avoid parity anomaly)

$$\chi^m \equiv \psi^m + i\psi^{m+N/2} \, , \quad \bar{\chi}^m \equiv \bar{\psi}^m - i\bar{\psi}^{m+N/2} \, , \quad m = 1, \dots, N/2 \, , \tag{3.17}$$

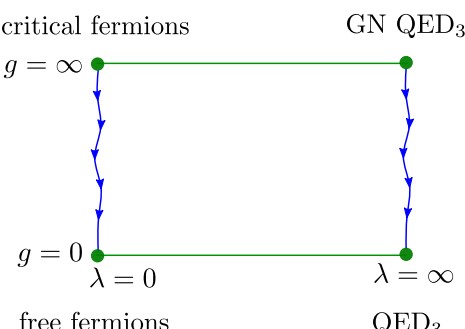

Figure 14: The four 3d fermionic CFTs connected via the interactions to the bulk scalar and the bulk gauge field. The horizontal direction represents the exactly marginal gauge coupling, while the vertical direction represents the RG flow triggered by the coupling to the bulk scalar.

so that the action is

$$S_N^f[g,\lambda] = \int_{y\geq 0} d^3x\,dy \left[ \frac{1}{2}(\partial_\mu \Phi)^2 + \frac{N}{4\lambda} F_{\mu\nu}F^{\mu\nu} \right] + \int_{y=0} d^3x \left[ \bar{\chi}^m \slashed{D}\chi^m + \frac{g}{\sqrt{N}}\,\Phi\,\bar{\chi}^m\chi^m \right],$$
(3.18)

where $D_a \equiv \partial_a + iA_a$. This theory interpolates between four different bulk/boundary decoupling limits, each of which contains a distinct 3d local CFT (see fig. 14)

- $N$ free Majorana fermions for $g=0$ and $\lambda=0$,

- the $O(N)$-symmetric Gross-Neveu model for $g=\infty$ and $\lambda=0$,

- the Gross-Neveu QED$_3$ with $N/2$ Dirac fermions, for $g=\infty$ and $\lambda=\infty$,

- critical QED$_3$ with $N/2$ Dirac fermions, for $g=0$ and $\lambda=\infty$.

Here by the Gross-Neveu QED$_3$ we mean the CFT obtained from the $O(N)$ Gross-Neveu model by gauging a $U(1)$ subgroup of the $O(N)$ symmetry and flowing to the IR, or equivalently a UV fixed point of critical QED$_3$ with $N/2$ Dirac fermion deformed by an irrelevant quartic interaction. While at the leading order both $g$ and $\lambda$ are marginal couplings, $\lambda$ remains exactly marginal to all orders while $g$ has a $\lambda$-dependent $\beta$ function. The Feynman rules for the theory $S_N^f[g,\lambda]$ are given in appendix A. The large $N$ propagator of $A_a$ between two points on the boundary is (up to gauge redundancy)

$$\langle A_a(p)A_b(-p)\rangle = \frac{1}{N}\frac{\lambda}{1+\lambda/32}\frac{\delta_{ab}}{|p|}.$$
(3.19)

### 3.4.1 Beta function and anomalous dimension of lowest singlet

Compared to the case without the gauge field, the $\beta$ function gets additional contributions from the diagrams depicted in fig. 15. We obtain

$$\beta_g = \beta_g|_{\lambda=0} - \frac{4}{3\pi^2 N}\frac{g\,\lambda(1-\lambda/16)}{(1+\lambda/32)^2} + \mathcal{O}(N^{-2}),$$
(3.20)

where the result for $\lambda=0$ is in eq. (3.3). Rewritten in terms of the variables $f_g \equiv \frac{g^2/16}{1+g^2/16}$ $\in [0,1]$ and $f_\lambda \equiv \frac{\lambda/32}{1+\lambda/32} \in [0,1]$ it gives

$$\beta_{f_g} = \frac{64}{3\pi^2 N}f_g(1-f_g)[f_g - 4f_\lambda(1-3f_\lambda)],$$
(3.21)

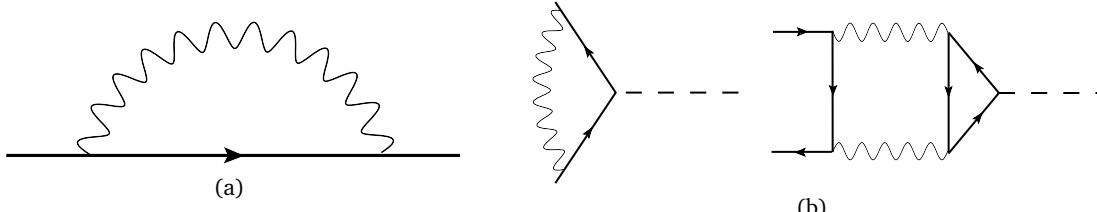

Figure 15: Diagrams that compute the photon contribution to the renormalization constants at order $1/N$. (a) gives the wavefunction renormalization of $\chi^m$ and (b) the renormalization of the $g$ vertex (permutations of external legs are omitted).

which is the same as the one in the bosonic theory written in the analogous variables - see eq. (2.46). Therefore, like in the bosonic case, there are three families of fixed points labeled by the value of the exactly marginal gauge coupling, two in which the bulk scalar is decoupled for $f_g = 0$ and $f_g = 1$, and one with the bulk scalar coupled for $f_g = 4f_\lambda(1-3f_\lambda)$. Since $4f_\lambda(1-3f_\lambda) \in [0,1]$ only for $f_\lambda = 0$ and $f_\lambda \in [1/3, 1/2]$, we cannot decouple the gauge field by taking $f_\lambda = 1$ to define a unitary interacting conformal boundary condition for the free scalar alone.

The modified Neumann boundary condition

$$\partial_y \Phi|_{y=0} = \frac{g}{\sqrt{N}} \bar{\chi}^m \chi^m \,, \tag{3.22}$$

together with the non-renormalization of the operator $\partial_y \Phi|_{y=0}$ allows us to fix the anomalous dimension of the bilinear operator $\bar{\chi}^m \chi^m$ in terms of the $\beta$ function

$$\gamma_{\bar{\chi}\chi} = \frac{\beta_g}{g} = \frac{2}{3\pi^2 N} \left( \frac{g^2}{1 + g^2/16} - \frac{2\lambda(1-\lambda/16)}{(1+\lambda/32)^2} \right) + \mathcal{O}(N^{-2}) \,. \tag{3.23}$$

For the family of fixed points with $g = 0$, this function interpolates between 0 in the free theory at $\lambda = 0$ and the limit $\lambda \to \infty$

$$\gamma_{\bar{\chi}\chi}|_{g=0,\lambda=\infty} = \frac{256}{3\pi^2 N} + \mathcal{O}(N^{-2}) \,, \tag{3.24}$$

that matches the result in QED$_3$ with $N/2$ Dirac fermions [36, 37]. In the limit $g \to \infty$ the operator $\bar{\chi}^m \chi^m$ vanishes, but we can still read off the dimension of the lowest singlet operator by looking at the limit of $-\gamma_{\bar{\chi}\chi}$. We argued for the analogous phenomenon in the case of the bosonic theory and we refer the reader to that discussion, below equation (2.53). In particular for $\lambda = 0$ we have

$$-\gamma_{\bar{\chi}\chi}|_{g=\infty,\lambda=0} = -\frac{32}{3\pi^2 N} + \mathcal{O}(N^{-2}) \,, \tag{3.25}$$

which matches the anomalous dimension of the HS field in the Gross-Neveu model [35], and in the limit $\lambda \to \infty$

$$-\gamma_{\bar{\chi}\chi}|_{g=\infty,\lambda=\infty} = -\frac{96}{\pi^2 N} + \mathcal{O}(N^{-2}) \,, \tag{3.26}$$

which matches the anomalous dimension of the HS field in the Gross-Neveu QED$_3$ [30, 38].

We observe that the anomalous dimensions in all four decoupling limits coincide with the corresponding ones in the bosonic theory. In fact the full function $\gamma_{\bar{\chi}\chi}$ in eq. (3.23) coincides with the bosonic analogue $\gamma_{z^\dagger z}$ in eq. (2.52) up to a redefinition $g \to 2g$ which does not affect the evaluation at $g = 0$ and $g = \infty$. Note that e.g. in the case of the $O(N)$ and Gross-Neveu CFT the anomalous dimension starts differing at order $\mathcal{O}(N^{-2})$ [39, 40], so also the equality of $\gamma_{\bar{\chi}\chi}$ and $\gamma_{z^\dagger z}$ up to rescaling of $g$ must fail at that order.

# 4 Outlook

Let us conclude by discussing some possible future directions.

- It would be interesting to study the $1/N$ corrections that lift the line of fixed points parametrized by $g$ and $g'$ from the point of view of the gravitational dual of the vector models. As we discussed, at the leading order we can describe holographically our setup as two conformally coupled scalars in $\text{AdS}_4$ with opposite boundary conditions, deformed by a mixed boundary condition that gives rise to the marginal coupling [9]. At order $1/N$ the scalar that belongs to the Vasiliev sector starts being subject to the interaction with the tower of higher spin gauge fields. It should be possible to reproduce the calculation of the $\beta$ functions from this point of view, perhaps by considering the one-loop correction to the bulk mixed two-point function of the two scalars, somewhat similarly to the one-loop corrections to two-point functions in Vasiliev theory considered in [41]. The complete off-shell formulation of the bulk theory recently proposed in [42, 43] might also be useful for this check. The holographic calculation of the $\beta$ function for a double-trace coupling induced by a bulk coupling was studied in [44].

- The vector models (projected to the singlet sector) belong to a continuous family of theories labeled by a parity-breaking deformation, that corresponds to coupling them to 3d non-abelian gauge fields for the $O(N)$ symmetry with large CS level $k$ and $N/k$ fixed [45–47]. It would be interesting to compute the corresponding one-parameter generalization of the RG that we studied in this paper, though it would be sensibly more involved technically. In particular it would be interesting to see if this additional parameter allows us to find real zeroes of the $\beta$ function in which the bulk scalar is not decoupled, therefore defining unitary interacting (parity-breaking) conformal boundary conditions for the free scalar.

- While all the claims of this paper are only valid in the large $N$ limit, it is tempting to speculate about their possible extension to finite $N$. The RG flows depicted in fig. 1 could still exist for finite $N$, though now the UV fixed point would be strongly coupled and the coupling $g'$ strongly relevant. On the other hand, both in the bosonic and in the fermionic theory the coupling $g$ on the free-vector-model side is classically marginal also for finite $N$, and it is possible to compute its $\beta$ function and the anomalous dimensions of operators in ordinary perturbation theory in $g$. The one-loop $\beta$ function for $g$ –both in the bosonic and in the fermionic theory– was calculated in [5]. Computing to some sufficiently high order one might then attempt an extrapolation to $g \to \infty$ and compare with the observables for the finite $N$ critical vector models. We note that at least in the bosonic theory the monotonicity of the hemisphere partition function is still satisfied by the RG in fig. 1 even for small $N$, if we use the estimate for the $F$ coefficient of the $O(N)$ model coming from $\epsilon$ expansion [48, 49].

- It would be interesting to compute the hemisphere partition function for the boundary theories studied in this paper and try to extrapolate the results along the RG flow. If such extrapolation is possible, it can be combined with the extrapolation in the gauge coupling introduced in [3] to compute the free energy of the gauged vector models starting from the free theory.

# Acknowledgements

We thank O. Aharony, R. Argurio, C. Behan, S. Benvenuti, M. Bertolini, D. Gaiotto, M. Serone and B. van Rees for interesting discussions. LD is partially supported by INFN Iniziativa Specifica ST&FI. LD also acknowledges support by the program "Rita Levi Montalcini" for young researchers. EL is supported by the Simons Foundation grant #488659 (Simons Collaboration on the non-perturbative bootstrap). PN is a Research Fellow of the F.R.S.-FNRS (Belgium).

# A   Feynman rules

## A.1   Dirichlet + $N$ free scalars

### A.1.1   Propagators

$$\partial_y\Phi \;\text{------}\; \partial_y\Phi \;=\; -\frac{|p|}{1+\frac{g^2}{4}} \qquad \varphi^I \;\text{------}\; \varphi^J \;=\; \frac{\delta^{IJ}}{p^2}$$

Figure 16: Large $N$ propagators of $\partial_y\Phi$ and $\varphi^I$, $I = 1,\dots,N$ for the Lagrangian (2.1).

### A.1.2   Vertices

$$\;=\; -\frac{2g}{\sqrt{N}}\delta^{IJ} \qquad\qquad \;=\; -\frac{48h}{N^2}\left(\delta^{IJ}\delta^{KL}\delta^{MN} + 14\,\text{perm}\right)$$

Figure 17: Cubic and sextic vertices for the Lagrangian (2.1). The sextic vertex is completely symmetric in the indices $I,J,K,L,M,N$.

## A.2   Neumann + $N$ critical scalars

### A.2.1   Propagators

$$\Phi \;\text{------}\; \Phi \;=\; \frac{1}{|p|} \qquad \varphi^I \;\text{------}\; \varphi^J \;=\; \frac{\delta^{IJ}}{p^2}$$

$$\sigma \;\cdots\cdots\cdots\; \sigma \;=\; -4|p|$$

Figure 18: Propagators of $\Phi$, $\sigma$ and $\varphi^I$, $I = 1,\dots,N$ for the Lagrangian (2.9) at $g' = 0$.

### A.2.2 Vertices

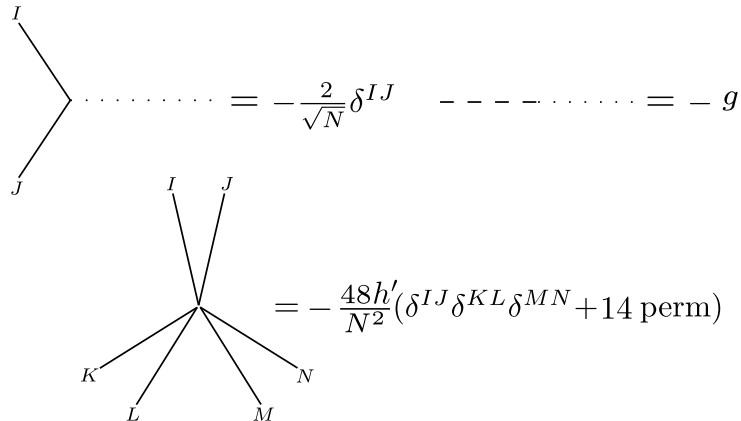

Figure 19: Vertices for the Lagrangian (2.9). The sextic vertex is completely symmetric in the indices $I, J, K, L, M, N$.

## A.3 Dirichlet scalar + Neumann gauge field + $N/2$ free complex scalars

### A.3.1 Propagators

$$\partial_y \Phi \;\text{-----}\; \partial_y \Phi \;=\; -\frac{|p|}{1+\frac{g^2}{4}} \qquad z^m \;\longrightarrow\; \bar{z}^n \;=\; \frac{\delta^{mn}}{p^2}$$

$$A_a \;\text{~~~~}\; A_b \;=\; \frac{1}{N}\,\frac{\lambda}{1+\frac{\lambda}{32}}\,\frac{\delta_{ab}}{|p|}$$

Figure 20: Large $N$ propagators of $\partial_y \Phi$, $z^m$, $m = 1, \ldots, N/2$ and $A_a$ for the Lagrangian (2.42).

### A.3.2 Vertices

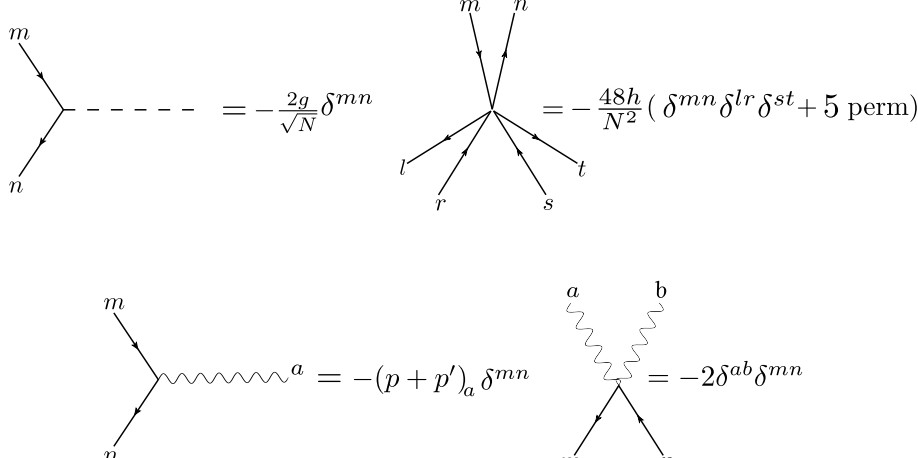

Figure 21: Vertices for the Lagrangian (2.42). The sextic vertex is completely symmetric in the indices $m, n, l$ and $r, s, t$. Momentum flows according to the arrows.

### A.4 Neumann + $N$ free Majorana fermions

#### A.4.1 Propagators

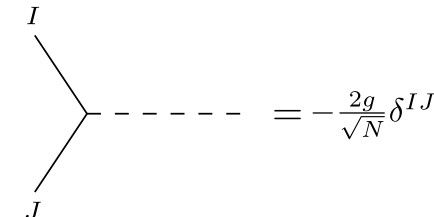

$$\Phi \;\text{-----------}\; \Phi \;=\; \frac{|p|^{-1}}{1+\frac{g^2}{16}} \qquad \psi^I \;\text{————————}\; \bar\psi^J \;=\; -\frac{i\!\!\not{p}\,\delta^{IJ}}{2p^2}$$

Figure 22: Large $N$ propagators of $\Phi$ and $\psi^I$, $I = 1,\dots,N$ for the Lagrangian (3.1).

#### A.4.2 Vertices



$$= -\frac{2g}{\sqrt{N}}\delta^{IJ}$$

Figure 23: Vertex for the Lagrangian (3.1).

### A.5 Dirichlet + $N$ critical Majorana fermions

#### A.5.1 Propagators

$$\partial_y\Phi \;\text{-------}\; \partial_y\Phi \;=\; -|p| \qquad \psi^I \;\text{————————}\; \bar\psi^J \;=\; -\frac{i\!\!\not{p}\,\delta^{IJ}}{2p^2}$$

$$\sigma \;\text{.................}\; \sigma \;=\; \frac{16}{|p|}$$

Figure 24: Propagators of $\partial_y\Phi$, $\sigma$ and $\psi^I$, $I = 1,\dots,N$ for the Lagrangian (3.5) at $g' = 0$.

#### A.5.2 Vertices

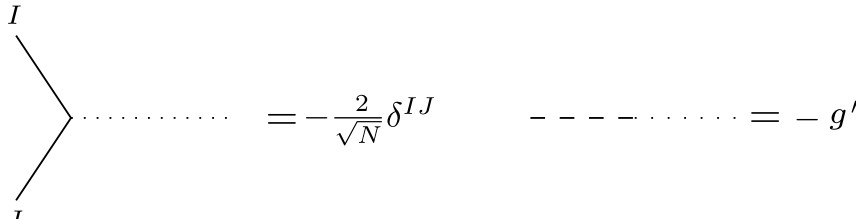

$$= -\frac{2}{\sqrt{N}}\delta^{IJ} \qquad\qquad = -g'$$

Figure 25: Vertices for the Lagrangian (3.5).

### A.6 Neumann scalar + Neumann gauge field + $N/2$ free Dirac fermions

#### A.6.1 Propagators

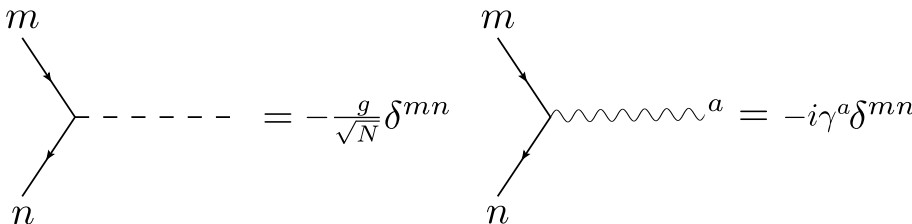

Figure 26: Large $N$ propagators of $\Phi$, $\chi^m$, $m = 1, \ldots, N/2$ and $A_a$ for the Lagrangian (3.18).

#### A.6.2 Vertices

$$
\begin{array}{l}
m \\
\\
\\
n
\end{array}
\quad = -\frac{g}{\sqrt{N}}\delta^{mn}
\qquad
\begin{array}{l}
m \\
\\
\\
n
\end{array}
{}^a = -i\gamma^a\delta^{mn}
$$

Figure 27: Vertices for the Lagrangian (3.18).

## B   Bosonic results with a bulk $\theta$ term

In this appendix we generalize the results of sections 2.4 by including a bulk $\theta$ term for the Maxwell field. This $\theta$ term allows us to engineer more general 3d abelian gauge theories at the boundary, when the bulk decouples [3,4]. The gauge field contribution to the action (2.42) is now

$$
S^{\text{Maxwell}}[\lambda, \gamma] = \int_{y \geq 0} d^3x \, dy \, \frac{N}{4\lambda} \left( F_{\mu\nu}F^{\mu\nu} + i\frac{\gamma}{2}\epsilon_{\mu\nu\rho\sigma}F^{\mu\nu}F^{\rho\sigma} \right). \tag{B.1}
$$

Our conventions are that $\epsilon_{abcy} = \epsilon_{abc}$, and $\gamma \equiv \frac{\lambda\theta}{4N\pi^2}$. Note that the large $N$ limit with $\lambda$ and $\gamma$ fixed corresponds to the limit in which the theta term is large and scales like $\mathcal{O}(N)$. The tree-level propagator in a generic $\xi$ gauge is

$$
\langle A_a(p)A_b(-p)\rangle^{(\text{tree})} = \frac{\lambda/N}{1+\gamma^2}\frac{1}{|p|}\left( \delta_{ab} - (1-\xi)\frac{p_a p_b}{|p|^2} + \gamma\epsilon_{abc}\frac{p^c}{|p|} \right). \tag{B.2}
$$

The exact propagator of the photon at the leading order in large $N$ (with $\gamma$ and $\lambda$ fixed), can be computed via the geometric resummation of bubbles as explained in subsection 2.4.1. The result in a generic $\xi$ gauge is

$$
\langle A_a(p)A_b(-p)\rangle = \frac{\lambda/N}{\gamma^2 + (1+\lambda/32)^2}\frac{1}{|p|}\left[ (1+\lambda/32)\left(\delta_{ab} - \frac{p_a p_b}{|p|^2}\right) + \gamma\epsilon_{abc}\frac{p^c}{|p|} \right] + \frac{\xi\lambda/N}{1+\gamma^2}\frac{p_a p_b}{|p|^3}. \tag{B.3}
$$

From now on we will fix $\xi$ such that the propagator does not contain any term proportional to $p_a p_b$. By repeating the computations of section 2.4 we find - compare to (2.44)

$$\delta Z_z = \delta Z_\varphi|_{\lambda=0} - \frac{5}{6\pi^2 N} \frac{\lambda(1+\lambda/32)}{\gamma^2 + (1+\lambda/32)^2} \log(\Lambda/\Lambda'),$$

$$\delta Z_g g = \delta Z_g g|_{\lambda=0} + \frac{1}{2\pi^2 N} \frac{g\lambda\left(\gamma^2(1+9\lambda/32) + (1+\lambda/32)^2(1-7\lambda/32)\right)}{\left(\gamma^2 + (1+\lambda/32)^2\right)^2} \log(\Lambda/\Lambda'),$$

$$\delta Z_h h = \delta Z_h h|_{\lambda=0} + \frac{1}{6\pi^2 N} \frac{(1+\lambda/32)\lambda}{(\gamma^2 + (1+\lambda/32)^2)} \left[ 9h\left(1 - \frac{\lambda\left(-\gamma^2 + (1+\lambda/32)^2\right)}{4(1+\lambda/32)(\gamma^2 + (1+\lambda/32)^2)}\right) \right.$$

$$\left. + \frac{\lambda^2\left(-3\gamma^2 + (1+\lambda/32)^2\right)}{(\gamma^2 + (1+\lambda/32)^2)^2} - \frac{6g^4}{(1+g^2/4)^2} \right] \log(\Lambda/\Lambda'). \tag{B.4}$$

From the above expression we obtain the $\beta$ functions for $f_g$ and $h$. In terms of the compactified variables $f_g = \frac{g^2/4}{1+g^2/4} \in [0,1]$, $f_\lambda = \frac{\lambda/32}{1+\lambda/32} \in [0,1]$ and $f_\gamma \equiv \frac{\gamma^2}{1+\gamma^2} \in [0,1]$ we find - compare to (2.46)

$$\beta_{f_g} = \frac{64 f_g(1-f_g)}{3\pi^2 N} \left( f_g - \frac{4(1-f_\gamma)(3f_\lambda+1)f_\lambda}{1+f_\gamma(f_\lambda-2)f_\lambda} + \frac{24(f_\gamma-1)^2 f_\lambda^2}{(1+f_\gamma(f_\lambda-2)f_\lambda)^2} \right),$$

$$\beta_h = \beta_h|_{f_\lambda=0} + \frac{2^7}{\pi^2 N} \frac{(f_\gamma-1)f_\lambda}{f_\gamma(f_\lambda-2)f_\lambda+1}$$

$$\times \left( h - 4f_g^2 + 3hf_\lambda + \frac{2(f_\gamma-1)f_\lambda(3h+64f_\lambda)}{1+f_\gamma(f_\lambda-2)f_\lambda} + \frac{2^9}{3} \frac{f_\lambda^2(f_\gamma-1)^2}{(1+f_\gamma(f_\lambda-2)f_\lambda)^2} \right). \tag{B.5}$$

There is an interesting two-parameter family of zeros for $\beta_{f_g}$ at

$$f_g = \frac{4(1-f_\gamma)(3f_\lambda+1)f_\lambda}{1+f_\gamma(f_\lambda-2)f_\lambda} - \frac{24(f_\gamma-1)^2 f_\lambda^2}{(1+f_\gamma(f_\lambda-2)f_\lambda)^2}. \tag{B.6}$$

To see which of the $f_\lambda$ and $f_\gamma$ above may correspond to unitary theories we need to impose that $f_g \in [0,1]$ and that, in turn, they correspond to real and positive zeros of $\beta_h$. The region such that $f_g \in [0,1]$ is shown in fig. 28. It is interesting to note that turning on $\gamma$ allows one to (just barely) get to the decoupling limit for the bulk gauge field, i.e. $f_\lambda = 1$, with $f_g \in [0,1]$ and not necessarily $= 0,1$. The decoupling point $f_\gamma = f_\lambda = 1$ lies on a corner of the unitary region in fig. 28, namely

$$f_\gamma = 1 - \alpha(f_\lambda-1)^2 + \mathcal{O}((f_\lambda-1)^3), \quad \alpha \in [0, (7-2\sqrt{10})/9] \vee \alpha \in [(7+2\sqrt{10})/9, 2], \tag{B.7}$$

and as we vary the parameter $\alpha$ within these intervals, $f_g$ attains all possible values in the interval $[0,1]$. One can check that also the $\beta$ function of the sextic coupling admits real and positive zeroes in this range. Therefore these points correspond to unitary and interacting conformal boundary conditions for the free scalar.

As an interesting special case, from our computation we can obtain the $\beta$ function of the sextic coupling in tricritical bosonic QED$_3$ with a Chern-Simons level $k$ for the $U(1)$ gauge group, in the limit of large $N$ and large $k$ with a fixed ratio $\kappa = k/N$. To this end we simply plug $g = 0$, $\gamma = \frac{\lambda\kappa}{2\pi}$ and take $\lambda \to \infty$ with $\kappa$ fixed, obtaining

$$\beta_h = \frac{1}{\pi^2 N} \left( -\frac{9}{32}h^3 + 9h^2 - \frac{2h\left(\frac{\kappa^2}{\pi^2} - \frac{1}{512}\right)}{\left(\frac{\kappa^2}{\pi^2} + \frac{1}{256}\right)^2} + \frac{\frac{\kappa^2}{\pi^2} - \frac{1}{768}}{4\left(\frac{\kappa^2}{\pi^2} + \frac{1}{256}\right)^3} \right). \tag{B.8}$$

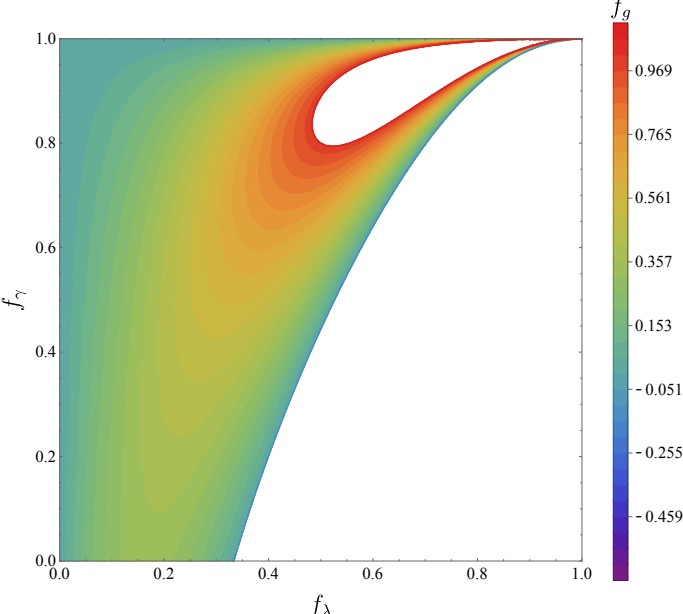

Figure 28: The region of $(f_\lambda, f_\gamma)$ such that $f_g \in [0,1]$. The colors represent different values of $f_g$ in this region. The blue lower bound $f_g(f_\lambda, f_\gamma) = 0$ and the red upper bound $f_g(f_\lambda, f_\gamma) = 1$ meet exactly at $f_\lambda = 1$. The upper left corner above is the region where only a large $\theta$ term is left in the bulk. In the upper right corner, the 4d Maxwell field decouples from the boundary and we are left with 3d abelian gauge fields with a large Chern-Simons level.

For $\kappa = 0$ this coincides with the result (2.47) at large $N$ with no CS term, while in the limit $\kappa \to \infty$ it gives the result for the large $k$, fixed $N$ perturbation theory, and it can be matched with the $\mathcal{O}(N^1)$ terms in the two-loop calculation of [50].[10] In fact the interacting boundary condition for the free scalar field found above can be also understood without invoking a bulk gauge field, simply arising from coupling the scalar with Dirichlet boundary condition to tricritical QED$_3$ at large $N$ and large $k$, via the interaction term $g \partial_y \Phi z^\dagger z$. The parameter $\alpha$ above corresponds to the parameter $\kappa$, i.e. $\alpha = \frac{\pi^2}{256\kappa^2}$. The $\beta$ function of the sextic coupling in Chern-Simons theories was considered recently in the case in which the rank of the gauge group is taken to be large in [51], and for $\mathcal{N} = 1$ supersymmetric theories in [52].

Also in this case we can further check our calculation of $\beta_h$ and $\beta_g$ by checking that in the limit (2.23) (recall that $g' = 1/g$ and $h' = h$) we obtain the $\beta$ function (2.22) of the cubic deformation $\Phi^3$ of the Neumann boundary condition, which indeed works for any value of $\lambda$ and $\gamma$.

## C  Dimensional regularization

In this work we have used the cutoff regularization in order to renormalize the different theories and compute their RG functions. It is instructive to check our results using dimensional regularization, i.e. taking the boundary dimension to be $d = 3 - \epsilon$ with fixed co-dimension

---

[10]Note that in [50] there are also fermions coupled to the CS gauge field, and they give the same contribution as the scalars when they run inside the bubble diagram that corrects the photon propagator. Therefore in order to reproduce their result we actually need to multiply by a factor of 2 the diagrams that in the limit $\kappa \to \infty$ contain one such bubble. Moreover, while the authors of [50] claim that they are omitting a factor $(64\pi^2)^{-1}$ in their RG functions, we find that actually the omitted overall factor in $\beta_h$ is $(16\pi^2)^{-1}$.

one.

As a paradigmatic example, we compute the $\beta$ function of the bulk/boundary coupling $g$ in the theory of the free bulk scalar field $\Phi$ with Dirichlet boundary condition coupled to $N$ free boundary scalars $\varphi^I$. We can set the sextic coupling $h$ to zero, since it does not enter in the renormalization of $g$. The bare action is

$$S_D^b[g,0] = \int_{y \geq 0} d^d x \, dy \, \frac{1}{2}(\partial_\mu \Phi_0)^2 + \int_{y=0} d^d x \left[ \frac{1}{2}(\partial_a \varphi_0^I)^2 + \frac{g_0}{\sqrt{N}} \partial_y \Phi_0 \, \varphi_0^I \varphi_0^I \right], \qquad \text{(C.1)}$$

where the bare coupling $g_0$ has mass dimension $\epsilon/2$. Recall the tree-level propagators and the vertex (see appendix A.1)

$$\Delta_\varphi^{(0)} \delta^{IJ} \equiv \langle \varphi^I(p) \varphi^J(-p) \rangle^{\text{(tree)}} = \frac{\delta^{IJ}}{p^2}, \qquad \langle \partial_y \Phi(p) \partial_y \Phi(-p) \rangle^{\text{(tree)}} = -|p|,$$

$$V^{\text{(tree)}} \delta^{IJ} = -\frac{2g_0}{\sqrt{N}} \delta^{IJ}. \qquad \text{(C.2)}$$

In order to renormalize the theory, we introduce the renormalized fields $\Phi$ and $\varphi^I$

$$\Phi_0 = \Phi, \qquad \varphi_0^I = \sqrt{Z_\varphi} \varphi^I, \qquad \text{(C.3)}$$

where we have used that $Z_\Phi = 1$, as required by locality of the theory. We introduce a sliding scale $\mu$ to define the renormalized dimensionless coupling constant $g$

$$g_0 Z_\varphi = g Z_g \mu^{\epsilon/2}. \qquad \text{(C.4)}$$

Plugging in the original action, the boundary terms in (C.1) become ($\delta_i = Z_i - 1$)

$$\int_{y=0} d^d x \left[ \frac{1}{2}(\partial_a \varphi^I)^2 + \frac{g \mu^{\epsilon/2}}{\sqrt{N}} \partial_y \Phi \, \varphi^I \varphi^I + \frac{\delta_\varphi}{2}(\partial_a \varphi^I)^2 + \frac{\delta_g g \mu^{\epsilon/2}}{\sqrt{N}} \partial_y \Phi \, \varphi^I \varphi^I \right]. \qquad \text{(C.5)}$$

The conterterms $\delta_i$ are fixed in order to cancel the UV divergences arising from the loop corrections in the renormalized fields, so that the theory (C.5) is UV finite.

Let us find the relation between the $\delta_i$ and the Feynman diagrams correcting the scalar propagator and the vertex (without the counterterm contribution), which we define to be $\Sigma_\varphi p^2 \delta^{IJ}$ and $2\nu g \mu^{\epsilon/2} \delta^{IJ}/\sqrt{N}$, respectively. At the leading order in the large $N$ expansion (since we will show that $\delta_i$ are $\mathcal{O}(N^{-1})$) the renormalized scalar propagator and the vertex are

$$\Delta_\varphi = \Delta_\varphi^{(0)} + \Delta_\varphi^{(0)} \left( \Sigma_\varphi p^2 - \delta_\varphi p^2 \right) \Delta_\varphi^{(0)}, \qquad V = \frac{2\mu^{\epsilon/2}}{\sqrt{N}} (-g + \nu g - \delta_g g). \qquad \text{(C.6)}$$

To cancel the divergences, we need that

$$Z_\varphi = 1 + \delta_\varphi = 1 + \Sigma_\varphi|_\infty, \qquad Z_g = 1 + \delta_g = 1 + \nu|_\infty, \qquad \text{(C.7)}$$

where the subscript indicates the divergent part when $\epsilon = 0$.

We can extract the $\beta$ function by requiring that the bare coupling constant $g_0$ in (C.4) does not depend on the sliding scale $\mu$

$$0 = \mu \frac{dg_0}{d\mu} = \mu^{\epsilon/2} \left( \frac{\epsilon}{2} \frac{Z_g g}{Z_\varphi} + \beta(g) \frac{\partial}{\partial g} \left( \frac{Z_g g}{Z_\varphi} \right) \right). \qquad \text{(C.8)}$$

This implies

$$\beta(g) = -\frac{\epsilon}{2} \left( \frac{\partial}{\partial g} \log \left( \frac{Z_g g}{Z_\varphi} \right) \right)^{-1} = -\frac{\epsilon}{2} g + \frac{\epsilon}{2} g^2 \frac{\partial}{\partial g} (\delta_g - \delta_\varphi) + \mathcal{O}(N^{-2}). \qquad \text{(C.9)}$$

The first term on the r.h.s. above is the classical contribution to the $\beta$ function. The second term is the quantum contribution, which survives when $\epsilon = 0$ since its $\epsilon$ dependence is canceled by the $1/\epsilon$ poles of $\delta_i$.

Next, we want to compute $\delta_\varphi$ and $\delta_g$, at order $\mathcal{O}(N^{-1})$ in the large $N$ limit (with $g$ fixed). To this end, we need the boundary propagator of $\partial_y \Phi$ for generic boundary dimension. This is obtained after resumming the scalar bubbles connected by the tree-level propagator of $\partial_y \Phi$, as in fig. 2. A straightforward computation in $d$ dimensions gives

$$\langle \partial_y \Phi(p) \partial_y \Phi(-p) \rangle = -|p| \sum_{k=0}^{\infty} \left( -C_d \left( \frac{\mu}{|p|} \right)^{3-d} \frac{g^2}{4} \right)^k = \frac{-|p|}{1 + C_d \left( \frac{\mu}{|p|} \right)^\epsilon g^2/4}, \tag{C.10}$$

where

$$C_d = \frac{2^{6-d} \sqrt{\pi} \, \Gamma(2-d/2)\Gamma(d/2-1)}{(4\pi)^{d/2} \Gamma(\frac{d-1}{2})}. \tag{C.11}$$

Note that this propagator is finite at $d = 3$, its expression being the one we used in the cutoff scheme. However, in order to compute correctly the divergent part of the counterterms, one should use the full $d$-dimensional form of the propagator and only at the end extract the $1/\epsilon$ pole by taking the limit $d = 3$. In fact, each $k$ term of the series in (C.10) gives a contribution to the pole in the counterterms. Hence, when computing the divergent Feynman diagrams, it is useful to write the boundary propagator of $\partial_y \Phi$ as an explicit geometric sum, exchange it with the loop integral, compute its divergence with the usual techniques and then perform the sum over $k$ to get the final result.

The Feynman diagram correcting the scalar propagator (see fig. 3 (a)) reads

$$\Sigma_\varphi p^2 = \left( -\frac{2g\mu^{\epsilon/2}}{\sqrt{N}} \right)^2 \int \frac{d^d q}{(2\pi)^d} \frac{-|q|}{(q+p)^2} \sum_{k=0}^{\infty} \left( -C_d \left( \frac{\mu}{|q|} \right)^{3-d} \frac{g^2}{4} \right)^k, \tag{C.12}$$

which gives

$$\delta_\varphi = \Sigma_\varphi|_\infty = -\frac{2g^2}{3\pi^2 N \epsilon} \sum_{k=0}^{\infty} \frac{1}{k+1} \left( -\frac{g^2}{4} \right)^k = -\frac{8}{3\pi^2 N \epsilon} \log\left( 1 + \frac{g^2}{4} \right). \tag{C.13}$$

The Feynman diagram correcting the vertex (see fig. 3 (b)) reads

$$\frac{2v g \mu^{\epsilon/2}}{\sqrt{N}} = \left( -\frac{2g\mu^{\epsilon/2}}{\sqrt{N}} \right)^3 \int \frac{d^d q}{(2\pi)^d} \frac{-|q|}{q^4} \sum_{k=0}^{\infty} \left( -C_d \left( \frac{\mu}{q} \right)^{3-d} \frac{g^2}{4} \right)^k, \tag{C.14}$$

which gives

$$\delta_g = v|_\infty = \frac{2g^2}{\pi^2 N \epsilon} \sum_{k=0}^{\infty} \frac{1}{k+1} \left( -\frac{g^2}{4} \right)^k = \frac{8}{\pi^2 N \epsilon} \log\left( 1 + \frac{g^2}{4} \right). \tag{C.15}$$

We can finally compute the anomalous dimension of the scalar field at large $N$

$$\gamma_\varphi = \frac{1}{2} \frac{d \log Z_\varphi}{d \log \mu} = \frac{1}{2} \beta(g) \frac{\partial \delta_\varphi}{\partial g} = \frac{1}{3\pi^2 N} \frac{g^2}{1 + g^2/4}, \tag{C.16}$$

and the $\beta$ function of the coupling

$$\beta(g) = -\frac{\epsilon}{2} g + \frac{\epsilon}{2} g^2 \frac{\partial}{\partial g} (\delta_g - \delta_\varphi) = -\frac{\epsilon}{2} g + \frac{8}{3\pi^2 N} \frac{g^3}{1 + g^2/4}. \tag{C.17}$$

Putting $\epsilon = 0$ we get the same result as in the main body of our work. Notice that the counterterms are given by different functions of the coupling $g$ in the two regularization schemes (nicely the first term in a series expansion around $g = 0$ matches in the two cases upon identifying $1/\epsilon \leftrightarrow \log(\Lambda/\Lambda')$), but they yield the same physical result.

Finally, let us derive using dimensional regularization the relation between $\beta_g$, the anomalous dimension of the Hubbard-Stratonovich field $\sigma$ in the $S_N^b[g', h']$ theory and the anomalous dimension of $\varphi^I \varphi^I$. This relation was derived e.g. in subsection 2.4, using the cutoff scheme. Following the duality map of table 1, we have that

$$\sigma_0 = g_0 \partial_y \Phi_0 \,, \tag{C.18}$$

whereas for the renormalized fields (since $[\sigma_0] = [\sigma] = 2$)

$$\sigma = \mu^{\epsilon/2} g \partial_y \Phi \,. \tag{C.19}$$

The non-renormalization of $\Phi_0$ implies $\sqrt{Z_\sigma} = Z_g Z_\varphi^{-1}$, which allows us to compute the anomalous dimension of $\sigma$ at large $N$ as

$$\gamma_\sigma = \frac{d \log \sqrt{Z_\sigma}}{d \log \mu} = \beta(g) \frac{\partial}{\partial g} \left( \log \left( \frac{Z_g g}{Z_\varphi} \right) - \log g \right) = -\frac{\epsilon}{2} - \frac{\beta(g)}{g} = -\frac{\beta(g)}{g} \bigg|_{d=3} \,. \tag{C.20}$$

Moreover, from the modified Dirichlet condition it follows immediately that $Z_{\varphi^2} = Z_\varphi Z_g^{-1}$, implying

$$\gamma_{\varphi^2} = -\gamma_\sigma = \frac{\beta(g)}{g} \bigg|_{d=3} \,. \tag{C.21}$$

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
