# Peer review of "d Large $N$ Vector Models at the Boundary"

_SciPost Physics, doi:SciPost Phys. 11, 050 (2021)_

## Round 2 · Referee Report · Anonymous (Referee 1) · 2021-4-1

Strengths

  1. The paper is very clearly written.

  2. The results match previous results in various limits, so are probably correct.

Weaknesses

  1. It's a bit disappointing that only the standard non-interacting conformal boundary conditions were found in the theories considered, tho this matches numerical evidence from a previous bootstrap study. The authors mention that generalizing their work to Chern-Simons matter theories could possibly give interacting boundary conditions, which would be very interesting.

Report

This paper presents results on various bosonic and fermionic vector models in 3d coupled to a free scalar in 4d, studied in the limit of many flavors. In particular, they find that the coupling of various theories come in dual pairs, which they prove using a path integral argument. The paper is well written and correct, and deserves to be published.

Requested changes

  1. When the authors consider the fermionic theory coupled to a U(1) gauge field, they should keep in mind that the parity anomaly requires that the number of complex 2 component fermions be even. This will not qualitatively change the main results tho.

  2. In the conclusion the authors say "due to to the presence of matrix-like degrees of freedom in the large N limit". When the number of colors and Chern-Simons level are both simultaneously large, the theory in fact still has vector degrees of freedom, which is in part why it is still believed to be dual to Vasiliev theory (with a parity breaking term).

  3. The authors consider generalizing their result to Chern-Simons matter theories with many colors. They should also consider adding a Chern-Simons term just to the U(1) gauge field, which is much simpler, and should give different boundary conditions for each integer value of the Chern-Simons level.

  4. The authors should also address monopole operators, which appear in QED3 in general. How are these operators affected by the 4d bulk theory? Is it clear how they would map across the duality? Similarly, it would be interesting to see how non-local operators like Wilson loops map across the duality.

---

## Round 2 · Referee Report · Anonymous (Referee 2) · 2021-7-2

Strengths

The authors provide a new and detailed analysis of an interesting quantum field theory with a boundary. The theory has a free scalar in the bulk that couples to either fermions or scalars on the boundary in a fundamental representation of O(N).

Weaknesses

The ratio of work required to information gained is large. A dozen pages of appendices provide details of the Feynman diagram calculations performed, while for the main characters in the paper no new interacting boundary conformal field theory is found. Instead, the bulk and boundary decouple at the fixed points, leading to well known interacting purely 3d CFTs. The one exception is their theory involving boundary scalars coupled to both a bulk scalar and photon, with a theta angle. In this case, in appendix B, they claim to find such interacting examples. I wonder if appendix B should have been the central focus of the paper. (Note that a bulk theta term is equivalent in this boundary context to a boundary Chern-Simons term.)

Report

There is a sense in which this paper can be considered as a follow-up of an observation in Witten's 2001 paper [9] of a duality involving two equal mass scalar fields in anti-de Sitter space with different boundary conditions. After a Weyl transformation and a replacement of one of the bulk fields with its boundary degrees of freedom, one finds the type of large N field theories studied in this paper. It is very satisfying to see all the details of this old observation worked out in this more familiar boundary QFT context.

Without hesitation, I recommend publication.

Requested changes

There were a couple of minor points:

In the introduction $\lambda$ is introduced without stating that it is the gauge coupling.

I lost the thread of the argument in the first bullet point on p 15. The authors state ``the operators $\Phi$ vanishes in the limit, as expected for a decoupled bulk free scalar with Neumann boundary condition''. Naively, I would want $\partial_\perp \Phi$ on the boundary to vanish for Neumann, and so I was not sure what was being said. It might help to reference some equations earlier in the draft, (2.24) and (2.25) for example.

---

## Round 3 · Referee Report · Anonymous (Referee 1) · 2021-7-8

Report

The updated paper looks good to me and is ready for publication.

---

## Round 3 · Author Response

Dear editor,

we would like to thank both referees for reading our manuscript and for their useful comments.

Before addressing their specific requests, we would like to reply to the comment from both referees about only having fixed points with bulk and boundary decoupled (though we bring to the attention of referee 1 that actually in the appendix B we do consider an example of a fixed point with bulk-boundary interactions). We certainly agree that interacting conformal boundary conditions are of prime interest. On the other hand we think that the interesting aspect of our models is that they provide examples of non-trivial boundary RG flows that can be followed from the UV to the IR, and admit dual descriptions from the two endpoints. In our opinion it is surprising and noteworthy that taking the limit of infinite bulk/boundary coupling one finds decoupling. As we tried to stress in the introduction, we find especially interesting that certain local interacting CFTs emerge from a boundary RG because they decouple from the bulk. Since we wanted to give more attention to this aspect we only studied the example with the theta term in the appendix B, and we decided to leave for the future a more thorough analysis of these interacting fixed points, including also the case with boundary fermions.

Let us now address one by one the more specific points the referees raised.

Reply to referee 1:

  • 1: We agree with the referee and we added a comment above equation (3.17);
  • 2: We agree that our terminology “matrix-like degrees of freedom” was poor given that the CS gauge fields do not propagate, we just meant to refer to the fact that the diagrams are considerably more involved than those of ungauged vector models, and indeed in a generic gauge one needs to resum all planar diagrams, like in a matrix-like large N limit. In any case to avoid imprecision we erased “due to the presence of matrix-like degrees of freedom”;
  • 3: We agree that this is an interesting direction. The theory with a CS level for the U(1) gauge field in 3d can be obtained by adding a bulk theta term for the 4d Maxwell field and going to infinite gauge coupling in the bulk (through EM duality). An initial study of this was performed in appendix B in the case of scalar matter on the boundary, as mentioned above. We did not consider the fermionic case because in the absence of parity there are more interactions to consider and the analysis changes substantially, so we decided to leave it for future work;
  • 4: Monopole operators in this setup would appear as endpoints of bulk ’t Hooft lines. Following the suggestion by the referee, we added this statement in the footnote 7, in which we were already discussing how operators charged under the gauge group can appear as endpoints of Wilson lines. We agree that analyzing such line operators is an interesting direction but we believe it goes beyond the scope of the present paper.

Reply to referee 2:

We added the definition of the coupling lambda in the introduction. Indeed there was a typo in the bullet of pag. 15 and we meant to say that the normal derivative of Phi goes to 0, we amended that.

Besides the requests from the referees, we made the following change: We realized that we had an inconsistency in the normalization of the kinetic term/propagator for the Majorana fermions compared to the quartic interaction, and therefore we had to correct various factors of 2, in their action and in their propagator. However this does not affect any physical result.

---

## Editorial Decision

published